# Causal Reinforcement Learning: A Survey

## Abstract

Reinforcement learning is an essential paradigm for solving sequential decision problems under uncertainty. Despite many remarkable achievements in recent decades, applying reinforcement learning methods in the real world remains challenging. One of the main obstacles is that reinforcement learning agents lack common knowledge of the world and must therefore learn from scratch through numerous interactions. They may also struggle to explain their decisions and generalize the learned knowledge. Causality, on the other hand, has a distinct advantage in that it can formalize knowledge and utilize structural invariance for efficient knowledge transfer. This has led to the emergence of causal reinforcement learning, a subfield of reinforcement learning that seeks to improve existing algorithms using structured and interpretable representations of the data generation process. In this survey, we comprehensively review the literature on causal reinforcement learning. We first introduce the basic concepts of causality and reinforcement learning, and then explain how causality can address core challenges in non-causal reinforcement learning. We categorize and systematically review existing causal reinforcement learning approaches based on their target problems and methodologies. Finally, we outline open issues and future directions in this emerging field.

## 1 Introduction

> *"All reasonings concerning matter of fact seem to be founded on the relation of cause and effect. By means of that relation alone we can go beyond the evidence of our memory and senses."*
>
> —David Hume, An Enquiry Concerning Human Understanding.

Humans have an innate ability to develop an understanding of causality from a young age (Wellman, 1992; Inagaki & Hatano, 1993; Koslowski & Masnick, 2002; Sobel & Sommerville, 2010). This level of understanding allows us to realize that changing certain things can cause others to happen; therefore, we can actively intervene in our environment to achieve desired goals or acquire new knowledge. Understanding cause and effect empowers us to explain behavior (Schult & Wellman, 1997), predict the future (Shultz, 1982), and even conduct counterfactual reasoning to reflect on past events (Harris et al., 1996). These abilities are essential to the development of human intelligence, laying the foundation for modern society and civilization as well as advancing science and technology.

For example, consider the story of humans battling scurvy (Pearl & Mackenzie, 2018), as illustrated in Figure 1. Scurvy once hampered human exploration of the world and claimed the lives of approximately 2 million sailors. After a long quest, humans discovered that consuming citrus fruits could prevent this terrible disease. Today, we know that the actual cause of scurvy is the lack of vitamin C, but back in the 19th century, this causal mechanism was unclear. People first believed acidity could cure the disease. However, heating the juice for purification destroyed the vitamin C content, rendering it ineffective against scurvy. People then believed acidity was only a placebo and that rotten meat was the cause of the disease. This misguided judgment took a heavy toll on Scott's Antarctica expedition. It was only when the causality of scurvy was fully understood that effective solutions to combat the disease were discovered. This example shows the importance of understanding causality in decision-making and the potentially disastrous consequences of ignoring it.

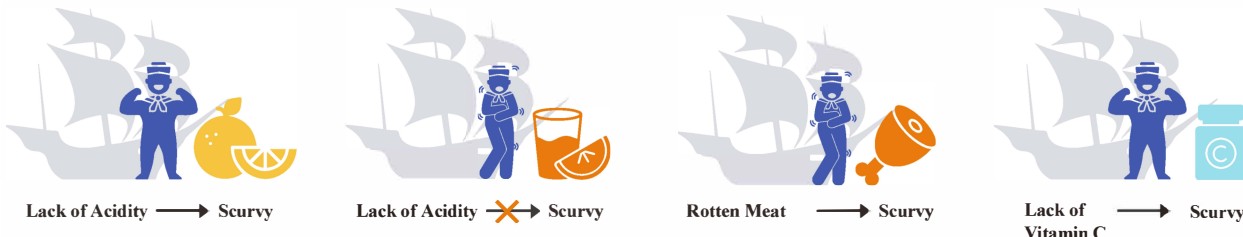

Figure 1: The journey to discover the cause of scurvy. At first, people thought that acidity could prevent the disease, which led to reckless decision-making, such as heating the juice for purification, which failed to address the root cause of the problem. Later, people started to believe that rotten meat was the cause of scurvy, which is a classic example of mistaking correlation for causality. Finally, it was discovered that the actual cause of scurvy is a deficiency in vitamin C. This breakthrough solved the mystery of scurvy and saved many lives.

Causality has long been a fundamental problem in machine learning. While pure data-driven methods [1] can capture correlations between variables, they often fail to interpret causal relationships (Pearl & Mackenzie, 2018). For instance, in the context of scurvy prediction, consuming rotten meat and getting scurvy might be strongly correlated but this does not imply causation. Vitamin C deficiency in the diet is the actual cause of scurvy. To understand causality, it is crucial to make and test assumptions about the data generation process. Causal inference has emerged as a powerful tool for addressing this issue by providing a mathematical framework for reasoning about causality in learning systems. (Schölkopf et al., 2021; Kaddour et al., 2022). In machine learning, causal inference has been applied in various fields, including computer vision (Lopez-Paz et al., 2017; Shen et al., 2018; Tang et al., 2020; Wang et al., 2020b), natural language processing (Wu et al., 2021; Jin et al., 2021; Feder et al., 2022), and recommender systems (Zheng et al., 2021; Zhang et al., 2021b; Gao et al., 2022). These results demonstrate that causality can significantly improve the robustness and interpretability of machine learning models and enable more effective knowledge transfer across domains.

Reinforcement learning (RL (Sutton & Barto, 2018) is a popular machine learning paradigm for decision-making problems. It involves actively intervening in the environment to learn from the outcome of certain behaviors. This property makes RL naturally connected to causality, as agents can make and test assumptions during the learning process. However, in most RL studies, agents are only allowed to intervene in action variables, making it challenging to fully understand the causal relationships that drive the underlying data generation process. This difficulty is further compounded in off-policy and offline settings due to the gap between the (possibly unknown) behavior policy and the target policy, and unobserved confounders that influence both action and outcome (Zhang et al., 2020b).

Causal RL is an emerging subfield of RL. It is an umbrella term for RL approaches that make explicit assumptions on the causal mechanisms behind the data to inform decision-making. For instance, imagine you were a captain in the 19th century embarking on a long voyage, trying to identify preventive measures for scurvy. When using traditional RL methods, massive exploratory behaviors are typically required, which can be costly and even fatal. In contrast, with causal RL, you would analyze the causal relationships and make wise assumptions beforehand. For example, based on the prior knowledge that food consumption is responsible for scurvy, you can avoid many meaningless and superstitious attempts, thereby increasing efficiency and safety. Furthermore, techniques from causal inference can help you discover that consuming rotten meat has no causal effect on getting scurvy, preventing you from drawing conclusions that fail to generalize. This example highlights the challenges facing traditional RL and the necessity of causal RL, which we will discuss in more detail in Section 3.

A range of settings has been explored in causal RL, from simple bandit problems to more complex Markov decision processes (MDPs), and from online to offline learning, with both model-free and model-based approaches. These studies have consistently demonstrated the superiority of causal RL over traditional meth-

---

[1]Pure data-driven methods refer to approaches that focus exclusively on summarizing or mining data, without considering the underlying mechanisms that govern its generation.

ods, providing more stable and effective solutions to decision-making problems. As causality plays a central role in human thought (Sloman, 2005; Sloman & Lagnado, 2015; Pearl & Mackenzie, 2018), one can expect causal RL to overcome many of the limitations of conventional methods and tackle new challenges in more complex scenarios. However, causal assumptions are usually encoded in different forms and for different purposes in previous research. The use of disparate terminologies and techniques makes it challenging to understand the essence, implications, and opportunities of causal RL, particularly for those new to causal inference and RL. This paper aims to provide a comprehensive survey of causal RL, integrating the various improvements and contributions made in this field. We establish connections between existing work based on the structural causal model (SCM) framework (Pearl, 2009a;b), which provides a systematic and principled manner to reason about causality.

Our main contributions to the field are as follows.

- Our survey of causal RL presents a comprehensive overview of the field, aligning existing research within the SCM framework. In particular, we introduce causal RL by answering three fundamental questions: What it is causal RL? Why does it need to be studied? And how do causal models improve existing RL approaches? We also present a clear and concise overview of the foundational concepts of causality research and RL. To the best of our knowledge, this is the first comprehensive survey of causal RL in the existing RL literature [2].

- We identify the bottleneck problems in RL that can be solved or improved by means of causality. We further propose a problem-oriented taxonomy. This taxonomy will help RL researchers gain a deeper understanding of the advantages of causality-aware methods and the opportunities for further research. On the other hand, RL practitioners can also benefit from this survey by identifying solutions to the challenges they face. Additionally, we compare and analyze existing causal reinforcement learning research based on their techniques and settings.

- We highlight major unresolved issues and promising research directions in causal RL, such as theoretical advances, benchmarks, and specific learning paradigms. These research topics are becoming increasingly and will help advance the use of RL in real-world applications. Therefore, establishing a common ground for discussing these valuable ideas in this emerging field is crucial and will facilitate its continued development and success.

## 2 Background

To better understand causal RL, an emerging field that combines the strengths of causality research and RL, we start by introducing the fundamentals of and some common concepts relevant to the two research areas.

### 2.1 A Brief Introduction to Causality

We first discuss how to use mathematical language to describe and study causality. In general, there are two primary frameworks that researchers use to formalize causality: SCM (structural causal model) Pearl (2009a); Glymour et al. (2016) and PO (potential outcome) (Rubin, 1974; Imbens & Rubin, 2015). We focus on the former in this paper because it provides a graphical methodology that can help researchers abstract and better understand the data generation process. It is noteworthy that these two frameworks are logically equivalent, and most assumptions are interchangeable.

**Definition 2.1** (Structural Causal Model)**.** An SCM is represented by a quadruple $(\mathcal{V}, \mathcal{U}, \mathcal{F}, P(\mathbf{U}))$, where

- $\mathcal{V} = \{V_1, V_2, \cdots, V_m\}$ is a set of endogenous variables that are of interest in a research problem,

- $\mathcal{U} = \{U_1, U_2, \cdots, U_n\}$ is a set of exogenous variables that represent the source of stochasticity in the model and are determined by external factors that are generally unobservable,

---

[2]We note that Schölkopf et al. (2021) and Kaddour et al. (2022) discussed causal RL alongside many other research subjects in their papers. The former mainly studied the causal representation learning problem, and the latter comprehensively investigated the field of causal machine learning. The present study, however, focuses on examining the literature on causal RL and provides a systematic review of the field.

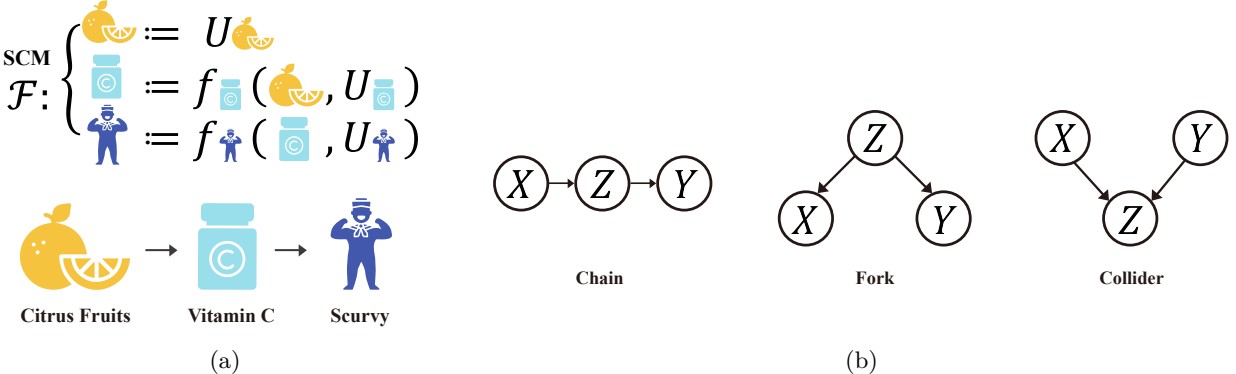

Figure 2: (a) A simplified version of the SCM and causal graph for the scurvy prediction problem. It includes three binary endogenous variables - consumption of citrus fruits, intake of vitamin C, and occurrence of scurvy - along with the relevant exogenous variables. (b) The three basic building blocks of causal graphs.

- $\mathcal{F} = \{f_1, f_2, \cdots, f_m\}$ is a set of structural equations that assign values to each of the variables in $\mathcal{V}$ such that $f_i$ maps $\mathbf{PA}(V_i) \cup U_i$ to $V_i$, where $\mathbf{PA}(V_i) \subseteq \mathcal{V} \backslash V_i$ and $U_i \subseteq \mathcal{U}$,

- $P(\mathbf{U})$ is the joint probability distribution of the exogenous variables in $\mathcal{U}$.

**Structural causal model**. SCM, as stated in definitionn 2.1, provides a rigorous framework for examining how relevant features of the world interact. Each structural equation $f_i \in \mathcal{F}$ specifies the value of an endogenous variable $V_i$ based on its direct causes $\mathbf{PA}(V_i) \cup U_i$. By defining these equations, we can establish the causal links between variables and mathematically characterize the underlying mechanisms of the data generation process. To generate samples from the joint distribution $P(\mathbf{V})$, we first sample the exogenous variables from $P(\mathbf{U})$, which represent the source of stochasticity in the model, and then evaluate the endogenous variables sequentially using the structural equations in $\mathcal{F}$. Once the values of the exogenous variables $\mathbf{U}$ are set, all endogenous variables $V_i \in \mathcal{V}$ are determined with perfect certainty. In particular, we primarily deal with Markovian models for which the exogenous variables are mutually independent [3].

**Causal graph**. Each SCM is associated with a causal graph $\mathcal{G} = \{\mathcal{V}, \mathcal{E}\}$, where nodes $\mathcal{V}$ represent endogenous variables and edges $\mathcal{E}$ represent causal relationships determined by the structural equations. Specifically, an edge $e_{ij} \in \mathcal{E}$ from node $V_j$ to node $V_i$ exists if the random variable $V_j \in \mathbf{PA}(V_i)$. Causal graphs typically omit exogenous variables unless they act as hidden confounders. In this case, a dashed line with bidirectional arrows is added to indicate that different endogenous variables are effects of the same confounder. Figure 2a illustrates the SCM of the scurvy problem (a simplified version) and the corresponding causal graph. Figure 2b introduces the three fundamental building blocks of the causal graph: chain, fork, and collider. These simple structures can be combined to create more complex data generation processes. In complicated scenarios where structural equations are unknown, causal graphs that encode our prior knowledge of causality as conditional independence may still be available, which are potentially useful in causal analysis.

**Intervention**. Intervention is a way of actively participating in the data generation process, rather than passively observing it. There are two types of interventions: hard interventions, where variables are directly set to constant values, and soft interventions, which modify the probability distribution of a variable while preserving some of its original dependencies. Many research questions involve predicting the effects of interventions. For example, finding a way to prevent scurvy is essentially about identifying effective interventions (through food or medicine) that lower the probability of getting scurvy. To differentiate from conditional probability, researchers introduced the do-operator, using $P(Y|\text{do}(X) = x)$ to denote the intervention probability, meaning the probability distribution of the outcome variable Y when variable $X$ is fixed to $x$. Figure 3a illustrates the difference between conditional and intervention probabilities. In particular,

---

[3]In general, the Markov assumption serves as a convention to determine the granularity of causal models. If $\mathbf{PA}(V_i)$ is too narrow, the Markov property will be lost due to disturbance terms that influence multiple variables simultaneously. This can be addressed by explicitly representing these terms as latent variables. For more information, see (Pearl, 2009b).

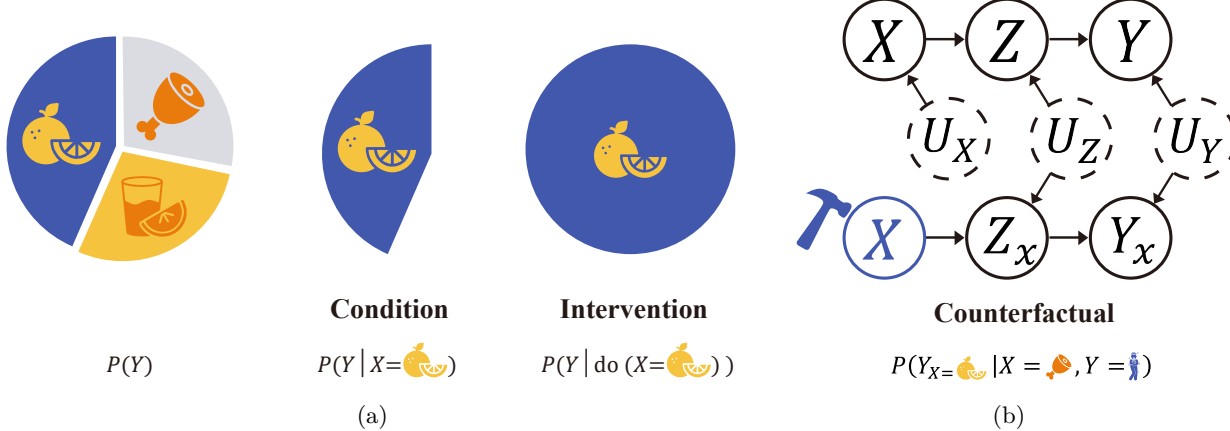

Figure 3: (a) An illustration of the difference between condition and intervention, in which $X$ is a discrete variable with three possible values: fresh citrus fruit, heated juice, or rotten meat. Marginal probabilities study all subgroups within a population, while conditional probabilities focus on a specific subgroup, like sailors who have consumed fresh citrus fruits. Intervention, on the other hand, examines the probability of scurvy by mandating that all sailors consume fresh citrus fruits. (b) An illustration of the counterfactual probability. Counterfactual probability explores events that occur in the imaginary world (the bottom network), such as asking whether sailors would be protected from scurvy if they had consumed enough citrus fruit, given that they consumed meat in the factual world (the upper network).

an intervention creates a new data distribution. From a causal perspective, a distribution shift occurs when data distributions originate from the same SCM but are subject to different interventions (Schölkopf et al., 2021; Thams, 2022).

**Counterfactual**. Counterfactual thinking is all about asking the "what if ...?" questions, such as "What if the scurvy patient had eaten enough citrus fruit? Would they stay healthy?". This type of thinking allows us to reflect on past events and consider how things might have been different if certain factors had been changed, helping us to learn from our experiences and make improvements.

In the context of SCMs, counterfactual variables are often denoted with a subscript, such as $Y_{X=x}$ (or $Y_x$ when there is no ambiguity) where $X$ and $Y$ are two sets of variables in $\mathcal{V}$. This notation helps researchers differentiate the counterfactual variables from the original variable $Y$. The key difference between $Y$ and $Y_x$ is that the latter is generated by a modified SCM with structural equations $\mathcal{F}_x = \{f_i : V_i \notin X\} \cup \{X = x\}$. Counterfactual reasoning, building on this formalism, aims to estimate probabilities such as $P(Y_{X=1}|X = 0, Y = 1)$. We can consider counterfactual reasoning as creating an imaginary world different from the factual one, whereas intervention only studies the factual one. See Figure 3b for a visual representation of counterfactual reasoning using the twin network method (Pearl, 2009b). The two networks represent the factual and counterfactual (imaginary) world respectively. They share the same structure, except the counterfactual one removes arrows pointing to the intervened variables. The two worlds are connected by the exogenous variables whose distribution is updated by the evidence. In particular, counterfactuals are assumed to satisfy consistency constraints such as $X = x \Rightarrow Y_x = Y$, meaning the counterfactual outcome matches its factual counterpart if the intervened variable is set to its actual value (Pearl, 2009b).

**Causal discovery and causal reasoning**. In the field of causal inference, two key areas of focus are causal discovery and causal reasoning. Causal discovery involves inferring the causal relationships between variables of interest (in other words, identifying the causal graph of the data generation process). Traditional approaches use conditional independence tests to infer causal relationships, and recently some studies have been conducted based on large datasets using deep learning techniques. Glymour et al. (2019) and Vowels et al. (2022) comprehensively survey the field of causal discovery.

As opposed to causal discovery, causal reasoning investigates how to estimate causal effects, such as intervention probability, given the causal model. Interventions involve actively manipulating the system or

environment, which can be costly and potentially dangerous (e.g., testing a new drug in medical experiments). Therefore, a core challenge of causal reasoning is how to translate causal effects into estimands that can be estimated from observational data using statistical methods. Given the causal graph, the identifiability of causal effects can be determined systematically through the use of do-calculus (Pearl, 1995).

**Product decomposition.** The causal Markov condition (Pearl, 2009b) states that for every Markovian causal model with a causal graph $\mathcal{G}$, the induced joint distribution $P(\mathbf{V})$ is Markov relative to $\mathcal{G}$. Specifically, a variable $V_i \in \mathcal{V}$ is independent of any variables that are not its direct causes or effects, given the set of all its direct causes $\mathrm{PA}(V_i)$ in $\mathcal{G}$. This property enables a structured decomposition along causal directions, which is referred to as the causal factorization (or the disentangled factorization) (Schölkopf et al., 2021):

$$P(\mathbf{V}) = \prod_{i=1}^{n} P(V_i | \mathrm{PA}(V_i)), \tag{1}$$

where the (conditional) probability distribution of the form $P(V_i | \mathrm{PA}(V_i))$ is known as the causal mechanism. Remark that equation 1 is not the only way to decompose the joint distribution $P(\mathbf{V})$. For instance, using the chain rule, we can derive $P(\mathbf{V}) = \prod_{i=1}^{n} P(V_i | V_1, \cdots, V_{i-1})$. However, equation 1 is the only approach that decomposes $P(\mathbf{V})$ as the product of causal mechanisms. To illustrate, let us consider Figure 2a, which depicts the data generation process of the scurvy prediction problem with a chain structure. We use the variables, $X$, $Z$, and $Y$ to represent citrus fruit consumption, vitamin C intake, and healthiness, respectively. The joint distribution $P(X, Y, Z)$ can be decomposed as $P(X)P(Z|X)P(Y|Z)$ or $P(Y)P(Z|Y)P(X|Y, Z)$. The former decomposition conforms to the causal graph of the given example, whereas the latter does not.

## 2.2 A Brief Introduction to Reinforcement Learning

Reinforcement learning studies sequential decision problems. Mathematically, we can formalize these problems as Markov decision processes.

**Definition 2.2** (Markov decision process). An MDP $\mathcal{M}$ is specified by a tuple $\{\mathcal{S}, \mathcal{A}, P, R, \mu_0, \gamma\}$, where

- $\mathcal{S}$ denotes the state space and $\mathcal{A}$ denotes the action space,

- $P : \mathcal{S} \times \mathcal{A} \times \mathcal{S} \to [0, 1]$ is the transition probability function that yields the probability of transitioning into the next states $s_{t+1}$ after taking an action $a_t$ at the current state $s_t$,

- $R : \mathcal{S} \times \mathcal{A} \to \mathbb{R}$ is the reward function that assigns the immediate reward for taking an action $a_t$ at state $s_t$,

- $\mu_0 : \mathcal{S} \to [0, 1]$ is the probability distribution that specifies the generation of the initial state, and

- $\gamma \in [0, 1]$ denotes the discount factor that accounts for how much future events lose their value as time passes.

**Markov decision processes.** In definition 2.2, the decision process starts by sampling an initial state $s_0$ with $\mu_0$. An agent takes responsive action using its policy $\pi$ (a function that maps a state to an action) and receives a reward from the environment assigned by $R$. The environment evolves to a new state following $P$; then, the agent senses the new state and repeats the interaction with the environment. The goal of an RL agent is to search for the optimal policy $\pi^*$ that maximizes the return (cumulative reward) $G_0$. In particular, at any timestep $t$, the return $G_t$ is defined as the sum of discounted future rewards, i.e., $G_t = \sum_{i=0}^{\infty} \gamma^i R_{t+i}$. A multi-armed bandit (MAB) is a special type of MDP problem that only considers a single-step decision. A partially observable Markov decision process (POMDP), on the other hand, generalizes the MDP by allowing for partial observability. While the system still operates on the basis of an MDP, the agent must make decisions based on limited information about the system state; for example, in a video game, a player may need to reason about the trajectory of the target object based on what is shown on the screen.

**Value functions.** Return $G_t$ can evaluate how good an action sequence is. However, when it comes to stochastic environments, the same action sequence can generate different trajectories, resulting in different

returns. In particular, a policy $\pi$ may also be stochastic. Given the current state, a policy $\pi$ can output a probability distribution over the action space. Thus, return $G_t$ is a random variable. To evaluate a policy, RL introduces the concept of value functions. There are two types of value functions: $V^\pi(s)$ denotes the expected return obtained by following the policy $\pi$ from state $s$; $Q^\pi(s, a)$ denotes the expected return obtained by performing action $a$ at state $s$ and following the policy $\pi$ thereafter. The optimal value functions that correspond to the optimal policy $\pi^*$ are denoted by $V^*(s)$ and $Q^*(s, a)$.

**Bellman equations**. By definition, $V^\pi(s) = \mathbb{E}_\pi[G_t|S_t = s]$ and $Q^\pi(s, a) = \mathbb{E}_\pi[G_t|S_t = s, A_t = a]$. These two types of value functions can be expressed in terms of one another. By expanding the return $G_t$, we can rewrite value functions in a recursive manner:

$$
\begin{aligned}
V^\pi(s) &= \sum_{a \in \mathcal{A}} \pi(a|s) \left( R(s, a) + \gamma \sum_{s' \in \mathcal{S}} P(s'|s, a) V^\pi(s') \right) \\
Q^\pi(s, a) &= R(s, a) + \gamma \sum_{s' \in \mathcal{S}} \sum_{a' \in \mathcal{A}} \pi(a'|s') Q^\pi(s', a').
\end{aligned}
\tag{2}
$$

When the timestep $t$ is not specified, $s$ and $s'$ are often used to refer to the states of two adjacent steps. The above equations are known as the Bellman expectation equations, which establish the connection between two adjacent steps. Similarly, the Bellman optimally equations relate the optimal value functions:

$$
\begin{aligned}
V^*(s) &= \max_{a \in \mathcal{A}} \left( R(s, a) + \gamma \sum_{s' \in \mathcal{S}} P(s'|s, a) V^*(s') \right) \\
Q^*(s, a) &= R(s, a) + \gamma \sum_{s' \in \mathcal{S}} P(s'|s, a) \max_{a' \in \mathcal{A}} Q^*(s', a').
\end{aligned}
\tag{3}
$$

When the environment (also referred to as the dynamic model or, simply, the model) is already known, the learning problem degenerates into a planning problem, which can be solved by dynamic programming based on the Bellman equation. However, RL focuses on the setting of unknown environments, i.e., the agents do not have complete knowledge in $P(s'|s, a)$ and $R(s, a)$, which aligns more closely with decision-making problems under uncertainty in the real world.

**Categorizing reinforcement learning methods**. There are several ways to categorize RL methods. One way is based on the components of the agent. Policy-based methods focus on optimizing an explicitly parameterized policy to maximize the return, while value-based methods use collected data to fit a value function and derive the policy implicitly from it. Actor-critic methods combine both of them, equipping an agent with a value function and a policy. Another way to classify RL methods is based on whether they use an environmental model. Model-based reinforcement learning (MBRL) typically uses a well-defined environmental model (such as AlphaGo (Silver et al., 2017)) or constructs one using the collected data. The model assists the agent in planning or generating additional training data, thus improving the learning process. Lastly, RL can be divided into on-policy, off-policy, and offline approaches based on how data is collected. On-policy RL only uses data from the current policy, while off-policy RL may involve the data collected by the previous policies. Offline RL disallows data collection, so the agent can only learn from a fixed dataset.

### 2.3 Causal Reinforcement Learning

Before formally defining causal RL, let us cast an RL problem into SCM. To do this, we consider the state, action, and reward at each step to be endogenous variables. The state transition and reward functions are then described as deterministic functions with independent exogenous variables, represented by the structural equations $\mathcal{F}$ in the SCM. The initial state can be considered an exogenous variable such that $S_0 \in \mathcal{U}$. This transformation is always possible using autoregressive uniformization (also known as the reparameterization trick) (Buesing et al., 2019), without imposing any extra constraints. It allows us to formally discuss causality in RL, including dealing with counterfactual queries that cannot be resolved by non-causal methods. Figure 4 presents an illustrative example of this transformation. In practice, states and actions may have high dimensionality, and the granularity of the causal model can be adjusted based on

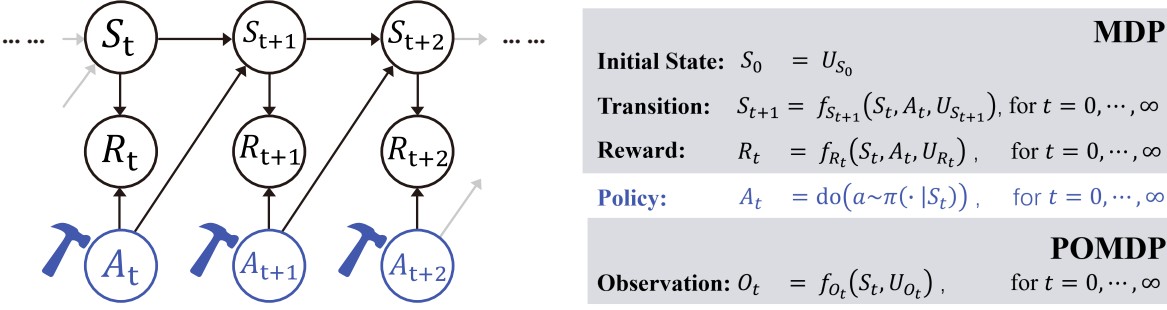

Figure 4: An illustrative example of casting an MDP or a POMDP problem into SCM. Actions are marked with hammers in the causal graph because they are intervened variables controlled by policies.

our prior knowledge. While the SCM representation allows us to reason about causality in decision-making problems and organize causal knowledge in a clear and reusable way, it does not constitute causal RL on its own. In this paper, we define causal RL as follows.

**Definition 2.3** (Causal reinforcement learning)**.** Causal RL is an umbrella term for RL approaches that focus on understanding and utilizing causal mechanisms of the underlying data generation process to inform decision-making, which involves incorporating additional causal assumptions into the learning process.

This definition highlights two key ways in which causal RL differs from non-causal RL. 1) It emphasizes a focus on causality, rather than relying on mere correlations or data patterns. To meet this goal, 2) it necessitates making additional assumptions regarding the causal relationships that arise in the decision-making problem.

In RL, the primary objective is to determine the policy $\pi$ that yields the highest expected return, rather than inferring the causal effect of an arbitrary intervention. The policy $\pi$ is a soft intervention that preserves the dependence of the action on the state, i.e., $\text{do}(a \sim \pi(\cdot|s))$. Different policies result in varying trajectory distributions. On-policy RL directly learns the causal effects of actions from interventional data, while off-policy and offline RL involve passively observing and learning from data collected by behavior policies, which may be easily affected by spurious correlations (Zhang et al., 2020b; Deng et al., 2021).

We note that there is a lack of clarity and coherence in the existing literature on causal RL, primarily because causal modeling is more of a mindset than a specific issue. Depending on the prior knowledge and research purpose, existing works have explored many different forms of causal modeling. Ideally, we can perfectly interpret the data generation process and answer various correlation, intervention and counterfactual queries, assuming we have access to the true SCM. This is usually not feasible in practice. However, we may assume structural knowledge of the SCM, i.e., the causal graph, which encodes conditional independence. With such structural knowledge and certain assumptions, we are already able to answer some interventional and counterfactual queries using observational data (Pearl & Mackenzie, 2018). Even if only part of the conditional independence is given, one can still greatly enhance RL (Seitzer et al., 2021). In pursuing efficient knowledge transfer, it is useful to examine the stable/invariant factors in the system, such as causal representation (causal variables that determine the data generation process), causal structure (causal graph), or causal mechanisms. These factors are probably unknown a priori and need to be learned from (possibly high-dimensional) raw data Sontakke et al. (2021); Huang et al. (2022a;b). Overall, the SCM formulation allows us to formally discuss causality in RL and retains the flexibility to accommodate different forms of causal modeling. In the next section, we will demonstrate how causal modeling not only improves our understanding of causal relationships in decision-making problems but also helps address problems in which non-causal approaches are insufficient.

# 3 Why Causality Is Important in Reinforcement Learning

Reinforcement learning has made remarkable advancements in the last decade, but it still carries some significant challenges. In this section, we summarize four major obstacles that have hindered the widespread use of RL algorithms in real-world applications. By taking a causal viewpoint, we can gain insight into these challenges and develop effective solutions. We scrutinize these challenges and explain why non-causal RL approaches may fail to address them.

## 3.1 Sample Efficiency in Reinforcement Learning

### 3.1.1 The Issue of Sample Efficiency in Reinforcement Learning

In RL, the data used for training is not provided beforehand. Unlike supervised and unsupervised learning methods that directly learn from a fixed dataset, an RL agent needs to actively gather new data to optimize its policy towards achieving the highest return. An effective RL algorithm should be able to master the optimal policy with as few experiences as possible (in other words, it must be sample-efficient). Current methods often require collecting millions of samples to succeed in even simple tasks, let alone more complicated environments and reward mechanisms. For example, AlphaGo Zero was trained over roughly $3 \times 10^7$ games of self-play (Silver et al., 2017); OpenAI's Rubik's Cube robot took nearly $10^4$ years of simulation experience (OpenAI et al., 2019). This inefficiency entails a high training cost and prevents the use of RL techniques for solving real-world decision-making problems that change rapidly. Therefore, the sample efficiency issue is a core challenge in RL, and developing sample-efficient RL algorithms that can save time and resources is essential.

### 3.1.2 Why Causal Modeling Helps Improve Sample Efficiency?

Sample efficiency is a critical concern in RL research (Kakade, 2003; Osband et al., 2013; Grande et al., 2014; Yu, 2018). Several factors affect it and abstraction is one of them. A well-abstracted problem eliminates unnecessary difficulties, leading to an efficient learning process. Some non-causal approaches (Jong & Stone, 2005; Zhang et al., 2022) realize abstraction by aggregating states that produce the same reward sequence. Despite reducing the complexity of the learning problem, they suffer from redundant dependencies and are vulnerable to spurious correlations. In contrast, causal relationships are generally sparse in reality (Pearl, 2009b; Huang et al., 2022a). See Figure 5 for a comparison between the two types of abstraction. By identifying the relevant features and necessary dependencies in the data generation process, causal modeling helps improve sample efficiency.

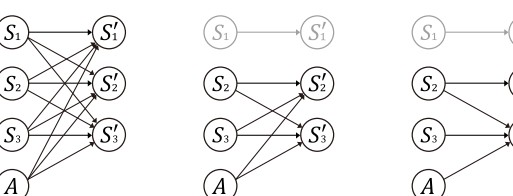

Figure 5: An illustration of a state transition between adjacent time steps. (Left) No abstraction. All variables are fully connected. (Middle) An irrelevant covariate $S_1$ is removed but the rest are still fully connected. (Right) Only the causal edges are preserved.

Designing effective exploration methods is another way to improve sample efficiency (Yang et al., 2022b). Some research has drawn inspiration from developmental psychology (Ryan & Deci, 2000; Barto, 2013) and used intrinsic motivation to motivate agents to explore unknown environments efficiently (Pathak et al., 2017; Burda et al., 2022). These approaches encourage agents to prioritize the exploration of regions of high epistemic uncertainty. They implicitly assume regions of equal uncertainty are equally important, which rarely holds in reality. Consider a robotic manipulation problem in which the agent needs to move an object to a target location. The state space (including information about the object and the robot arm) is vast, but most regions have limited information about the task, even if they show high uncertainty. Only the regions where the agent can causally influence the object (have a causal effect on the outcome, e.g., changing the momentum) are of interest. In cases like this, non-causal methods fail to learn efficiently while causal inference provides a principled way to estimate causal influence based on conditional independence (Seitzer et al., 2021), leading to better exploration.

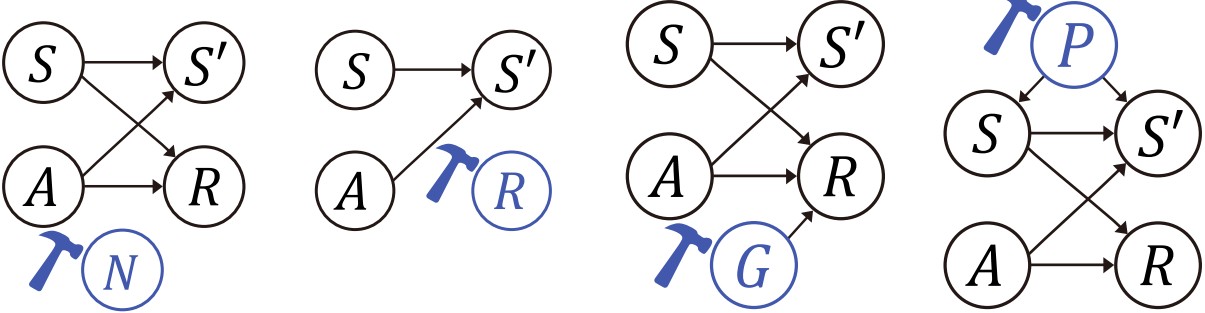

(a) Generalize to noise or ir-relevant variables.   (b) Generalize to different reward assignments.   (c) Generalize to different goals.   (d) Generalize to different physical properties.

Figure 6: Different types of generalization problems in reinforcement learning represented by causal graphs.

Model-based reinforcement learning (MBRL) (Wang et al., 2019; Luo et al., 2022) also helps improve sample efficiency. However, synthesizing trajectories de novo is highly challenging, especially for complex environmental dynamics. Compared to non-causal models built on correlations, causal models are more robust and enable counterfactual reasoning, which allows for generating samples based on real data. By explicitly considering the posterior distribution of exogenous variables, instead of using a fixed prior distribution such as the Gaussian distribution, causal RL helps generate higher-quality training data than non-causal approaches (Buesing et al., 2019).

### 3.2 Generalizability in Reinforcement Learning

#### 3.2.1 The Issue of Generalizability in Reinforcement Learning

Generalizability in RL is another major challenge to the deployment of RL algorithms in the real world. It refers to the ability of a trained policy to perform well in new, unseen situations (Kirk et al., 2022). Training and testing in the same environment has been a notorious problem in the RL community (Irpan, 2018). While people often expect RL to work reliably in different (but similar) environments or tasks, traditional RL algorithms are typically designed to solve a single MDP. They can easily overfit the environment, failing to adapt to minor changes. Even in the same environment, RL algorithms can produce widely varying results with different random seeds (Zhang et al., 2018a;b), indicating instability and overfitting. Lanctot et al. (2017) presented an example of overfitting in multi-agent scenarios in which a well-trained RL agent struggles to adapt when the adversary changes its strategy. A similar phenomenon was observed by Raghu et al. (2018). Additionally, the real world is non-stationary and constantly changing (Hamadanian et al., 2022), so a good RL algorithm must be robust in order to handle these changes. When the situation varies, agents should be able to transfer their skills effectively rather than starting from scratch.

#### 3.2.2 Why Causal Modeling Helps Improve Generalization and Facilitate Knowledge Transfer?

In RL, generalization involves different samples or environments. Kirk et al. (2022) proposed using contextual MDP (CMDP) (Hallak et al., 2015) to formalize the generalization problem in RL. CMDP is similar to a standard MDP, but it captures the variability in a set of environments or tasks determined by context, which can represent a variety of factors, such as random seeds, goals, colors, and difficulty of game levels.

Some previous studies have shown that data augmentation improves generalization (Lee et al., 2020; Wang et al., 2020a; Yarats et al., 2021), particularly for vision-based control. This process involves generating new data by randomly shifting, mixing, or perturbing observations, which makes the learned policy more resistant to irrelevant changes. Another common practice is domain randomization. In sim-to-real reinforcement learning, researchers randomized the parameters of simulators to facilitate adaptation to reality (Tobin et al., 2017; Peng et al., 2018). Additionally, some approaches have attempted to incorporate inductive bias

by designing special network structures to improve generalization performance (Kansky et al., 2017; Higgins et al., 2017; Zambaldi et al., 2019; Raileanu & Fergus, 2021).

Although these works have demonstrated empirical success, it is challenging to explain why certain techniques work better than others. This gap can hinder understanding of the keys to success in generalization and designing algorithms that are robust to different kinds of variations. To address this challenge, we need to identify the factors of change. From a causal perspective, changes in the joint distribution can be interpreted as an external intervention in the data generation process (Schölkopf et al., 2021; Thams, 2022). Figure 6 illustrates some examples of the generalization problems corresponding to different interventions. Previous methods simulate these interventions during the training phase by augmenting original data or randomizing certain attributes, allowing the model to learn from various domains. By carefully scrutinizing the causal relationships behind the data, we can gain a better understanding of the sources of generalization ability and provide a more logical explanation.

More importantly, by making explicit assumptions on what changes and what remains invariant, we can derive principled methods for effective knowledge transfer. To illustrate this point, let us recall the example discussed at the end of Figure 2, where we can use $X$, $Y$, and $Z$ to represent citrus fruit consumption, vitamin C intake, and occurrence of scurvy, respectively. Consider an intervention on fruit consumption (the variable $X$), one would have to retrain all modules in a non-causal factorization such as $P(X, Y, Z) = P(Y)P(Z|Y)P(X|Y, Z)$ due to the change in $P(X)$. In contrast, with the causal factorization $P(X, Y, Z) = P(X)P(Z|X)P(Y|Z)$, only $P(X)$ needs to be adjusted to fit the new domain. The intuition behind this is simple: changing fruit consumption $P(X)$ has no effect on the vitamin C content in a given fruit $P(Z|X = x)$ or the probability of getting scurvy given the amount of vitamin C intake $P(Y|Z = z)$. This property is known as *independent causal mechanism* (Schölkopf et al., 2021), indicating that the causal generation process consists of stable and autonomous modules (causal mechanisms) (Pearl, 2009b) such that changing one does not change the others. Based on this idea, the sparse mechanism shift hypothesis (Schölkopf et al., 2021; Perry et al., 2022) suggests that small changes in the data distribution generally reflect changes to only a subset of causal mechanisms, providing a principle for designing efficient machine learning algorithms and models for knowledge transfer. When the changed modules are task-independent (e.g., the background color in a robotic manipulation task), we can train a policy that focuses on invariance (Zhang et al., 2020a; Bica et al., 2021b), thereby achieving robust generalization.

### 3.3 Spurious Correlations in Reinforcement Learning

### 3.3.1 The Issue of Spurious Correlation in Reinforcement Learning

Learning decisions from data alone is insufficient because correlation does not necessarily imply causation. A spurious correlation is a relationship between two variables that appears to be causal but is actually caused by a third variable, bringing undesired bias to the learning problem. This phenomenon occurs widely in machine learning applications, with a few typical examples given below.

- In recommendation systems, both user behavior and preferences are influenced by conformity. If the recommender ignores conformity, it may overestimate a user's preference for certain items (Gao et al., 2022);

- In image classification, if dogs frequently appear with grass in the training set, the classifier may label an image of grass as a dog. This is because the model relies on the background (irrelevant factors) instead of the pixels corresponding to dogs (the actual cause) (Zhang et al., 2021a; Wang et al., 2021c).

- When determining the ranking of tweets, the use of gender icons in tweets is usually not causally related to the number of likes; their statistical correlation comes from the topic, as it influences both the choice of icon and the audience. Therefore, it is not appropriate to determine the ranking by gender icons (Feder et al., 2022).

If we want to apply RL in real-world scenarios, it is important to be mindful of spurious correlations, especially when the agent is working with biased data. For instance, when optimizing long-term user satisfaction

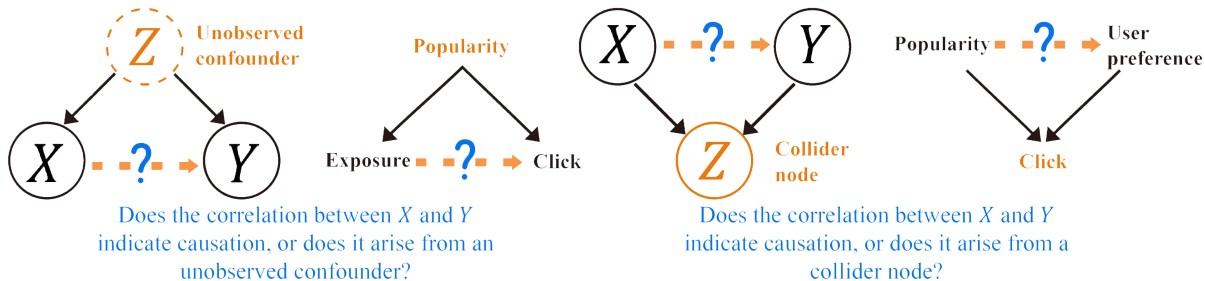

Figure 7: Causal graphs illustrating the two types of spurious correlations, with examples from real-world applications.

in multiple-round recommendations, there is often a spurious correlation between exposure and clicks in adjacent timesteps. This is because they are both influenced by item popularity. From another perspective, when we observe a click, it may depend on user preference or item popularity, which creates a spurious correlation between the two factors. In both scenarios, if the agent ignores causality, it will make incorrect predictions or decisions, such as only recommending popular items (a suboptimal policy for both the system and the user) and can lead to filter bubbles. In a nutshell, if the agent learns a spurious correlation between two variables, it may mistakenly believe that changing one will affect the other, even though there is no causal relationship between them in the underlying data generation process. This misunderstanding can lead to suboptimal or even harmful behavior in real-world decision-making problems.

### 3.3.2 Why Causal Modeling Helps Address Spurious Correlations?

The non-causal approaches lack a language for systematically discussing spurious correlations. From the causal perspective, spurious correlations arise when the data generation process involves unobserved confounders (common cause) or when a collider node (common effect) serves as the condition. The former leads to confounding bias, while the latter results in selection bias. See Figure 7 for a visual interpretation of these phenomena. Causal graphs enable us to trace the source of spurious correlations by closely scrutinizing the data generation process. To eliminate the bias induced by spurious correlations, it is necessary to make decisions regarding causality instead of statistical correlations. This is where causal reasoning comes in: It provides principled tools to analyze and deal with confounding and selective bias (Pearl, 2009b; Glymour et al., 2016), helping RL agents accurately estimate the causal effects in decision-making problems.

One may assume that on-policy RL is not susceptible to spurious correlations, as it directly learns causal effects of the form $p(r|s, \text{do}(a))$ from interventional data. However, the effect of actions on the outcome is only part of the picture upon opening the black box of the data generation process. It is often overlooked in RL how different covariates affect outcomes. For example, in personalized recommendation, the covariates can be divided into relevant and spurious features. If the clicked items always accompany specific values of the spurious features (e.g., item popularity and clickbait titles) in the training environment, then an agent may learn a policy based on these spurious features. In the test environment, the distribution of these spurious features may change, leading the agent to make incorrect decisions (Gao et al., 2022). This example shows spurious correlations widely exist in decision-making problems in the wild, hidden behind the data. Demystifying the causal relationships helps resolve the bias.

### 3.4 Considerations Beyond Return

In general, RL focuses on maximizing returns. However, as automated decision systems based on RL become a more prevalent feature of our daily lives, it is crucial to pay attention to how RL agents interact with people and how they may influence society.

### 3.4.1 Explainability in Reinforcement Learning

Explainability in RL refers to the ability to understand and interpret the decisions of an RL agent. It is important to both researchers and general users. Explanations reflect the knowledge learned by the agent, facilitate in-depth understanding, and allow researchers to participate efficiently in the design and continual optimization of an algorithm. In addition, explanations provide the internal logic of the decision-making process. When agents outperform humans, we can extract knowledge from the explanations to guide human practice in a specific domain. For general users, the explanation presents the reasons for the decision, thus deepening the user's understanding of intelligent agents and increasing the user's confidence in the agent.

### 3.4.2 Achieving Explainability through Causality

Explainable RL methods can be divided into two categories: post hoc and intrinsic approaches (Puiutta & Veith, 2020; Heuillet et al., 2021). The former provides explanations after execution, while the latter is inherently transparent. Post-hoc explanations are generally established based on correlations, such as the saliency map approach (Greydanus et al., 2018; Mott et al., 2019). As we mentioned earlier, conclusions based on correlations can be unreliable, failing to answer causal questions. On the other hand, intrinsic explanations can be achieved using easy-to-understand algorithms, such as linear regression or decision trees (Coppens et al., 2019). However, the limited model capability may be insufficient for explaining complicated behaviors (Puiutta & Veith, 2020).

Humans possess an innate and powerful ability to establish connections between different events through a "mental causal model (Sloman, 2005)". We frequently use causal language, such as "because," "therefore," and "if only," in our daily lives to facilitate communication and collaboration. Using causal models allows for natural and flexible explanations, as it does not require selecting specific algorithms or models. In RL, causality-based explainability provides stable support for the agent's decisions and helps us understand how the agent interprets the environment and task. When the agent makes mistakes, we can respond with more tailored solutions.

### 3.4.3 Fairness in Reinforcement Learning

As machine learning becomes more prevalent in our daily lives, stakeholders, such as business owners, general users, and policymakers are realizing the importance of fairness. This concept applies to any type of automated system and decision support system, including those based on RL. In particular, RL agents should strive to genuinely benefit people and promote social good rather than causing discrimination or harm against specific individuals or groups. In addition, fairness issues in the real world are often dynamic (Gajane et al., 2022), involving multiple decision points. For example, a hiring process is a typical sequential decision process, and the actions may have cumulative effects on fairness. Ignoring the dynamic nature of a system may lead to unintended unfairness (Liu et al., 2018; Creager et al., 2020; D'Amour et al., 2020).

### 3.4.4 Achieving Fairness through Causality

In legal cases, fairness or discrimination often involves a counterfactual statement (Pearl & Mackenzie, 2018). For example, in *Carson v. Bethlehem Steel Corporation* (1996) [4], the ruling made it clear that determining whether an individual or group would be treated differently if they changed only the sensitive attribute (e.g., sex, age, and race) while holding other factors constant is at the heart of discrimination issues. In machine learning, researchers often use demographic parity (Zafar et al., 2015; Wen et al., 2021), a metric built on the difference between the conditional distributions of different groups, to study fairness. However, as pointed out by Kusner et al. (2017) and Zhang & Bareinboim (2018), correlation-based metrics ignore the causal relationships behind the data generation process, resulting in an inaccurate measurement of fairness, which may increase discrimination in certain situations. As illustrated in Figure 3, conditional probabilities and counterfactual probabilities are significantly different. The SCM framework makes it clear that fairness issues require a comparison of the differences in causal effects between the factual and imagined worlds.

---

[4] https://caselaw.findlaw.com/us-7th-circuit/1304532.html

Therefore, using causal models is a principled way of studying fair reinforcement learning, which offers a new perspective compared to non-causal approaches.

### 3.4.5 Safety in Reinforcement Learning

Safety is a crucial aspect of RL (García & Fernández, 2015; Gu et al., 2022). RL agents may sometimes exhibit unexpected or unpredictable behavior, particularly when encountering new or unforeseen situations. This issue can pose a significant risk in safety-critical applications, such as healthcare or autonomous vehicles, where a single mistake could result in severe consequences. Additionally, RL agents may prioritize higher returns over their own safety, known as the agent safety problem (Fulton & Platzer, 2018; Beard & Baheri, 2022). A typical example is robotic control, where agents sacrifice their lifespan for a higher mission completion rate. In short, ensuring safety in RL is vital for preventing accidents or other harmful events in reality.

### 3.4.6 Achieving Safety through Causality

Researchers often use constrained MDPs (Altman, 1995; 1999) to model the safety issues in RL. Constrained MDPs extend MDPs by incorporating a constraint set that represents safety concerns. Accordingly, most relevant studies have focused on solving constrained optimization problems (Achiam et al., 2017; Chow et al., 2017), rarely considering causality. With the aid of causal models, we can formalize prior knowledge more effectively and obtain valuable insights, such as explanations, by analyzing the generation processes of unsafe states or actions. Causal models can assist researchers and experts in understanding the causes of unexpected outcomes and developing preventive solutions to avoid recurrence (Everitt et al., 2021), particularly when RL agents breach safety constraints. Additionally, causal models can be utilized for counterfactual policy evaluation, allowing for testing and identifying potential security issues before deploying an RL agent in real-world applications (Hart & Knoll, 2020). Overall, causal models help ensure RL techniques are used safely and responsibly, avoiding catastrophic consequences.

To summarize, we discussed several key challenges of RL in this section and scrutinized why causality helps solve or mitigate these challenges. Next, we review recent advances in causal RL.

## 4 Existing Work Relating to Causal Reinforcement Learning

In the previous section, we highlighted the significance of causal modeling in RL. It provides principles and insights to organize prior knowledge and assumptions, applying to a wide range of problems. This section reviews existing approaches to causal RL that attempt to address the four critical challenges outlined in section 3. We organize these approaches based on their problem settings and solution methods with the goal of better understanding their connections and relationships. To be self-contained, we offer a concise overview of the environment and task mentioned in this section in Appendix A.

### 4.1 Causal Reinforcement Learning for Addressing Sample Inefficienty

Causality offers some useful principles for designing sample-efficient RL algorithms. We can organize these principles into three lines of research: representation learning, directed exploration, and data augmentation. The representative works are shown in Table 1.

### 4.1.1 Representation Learning for Sample Efficiency

A good representation of the environment can be beneficial for sample-efficient RL. By providing a compact and informative representation of the environment, an RL agent can learn more effectively with fewer samples. This is because a good representation can help the agent identify important features of the environment and abstract away unnecessary details, allowing the agent to learn more generalizable policies and make better use of its experiences.

The research of Sontakke et al. (2021) involved clustering the trajectories generated from various environments with different physical properties and using the clustering outcomes as a causal representation. With

Table 1: Selected methods utilizing causality to optimize sample efficiency.

| Category | Paper | Technique | Environments or Tasks | Source |
|---|---|---|---|---|
| Representation Learning | Sontakke et al. (2021) | Causal representation learning | Manipulation (CausalWorld) | ICML |
| | Lee et al. (2021) | Intervention
Domain randomization | Manipulation (Isaac Gym) | ICRA |
| | Huang et al. (2022b) | Causal dynamics learning | Car Racing (OpenAI Gym)
VizDoom | ICML |
| | Wang et al. (2022) | Causal dynamics learning | Chemical
Manipulation (robosuite) | ICML |
| Directed Exploration | Seitzer et al. (2021) | Intervention | Manipulation (OpenAI) | NeurIPS |
| Data Augmentation | Buesing et al. (2019) | Counterfactual reasoning | Sokoban | ICLR |
| | Lu et al. (2020) | Counterfactual reasoning | Cart Pole (OpenAI Gym)
MIMIC-III | NeurIPS Workshop |
| | Pitis et al. (2020) | Counterfactual reasoning | Spriteworld
Pong (Roboschool)
Manipulation (OpenAI) | NeurIPS |
| | Zhu et al. (2021) | Counterfactual reasoning | Manipulation (CausalWorld) | OpenReview |

the state augmented by the causal representation, the learn policies exhibit outstanding zero-shot generalization ability, and require only a small number of training samples to converge in new environments. Causality also facilitates state abstraction. Lee et al. (2021) used interventions to identify the state variables that are important for successful task completion, reducing the dimensionality of the state space and simplifying the problem. When direct intervention on the states is infeasible, learning the causal dynamics model using collected data provides an alternative approach for state abstraction by utilizing the causal relationships between variables. Huang et al. (2022b) proposed using action-sufficient state representations, a minimal set of state variables containing sufficient decision-making information, to improve sample efficiency in RL. Wang et al. (2022), on the other hand, studied task-independent state abstraction which only omits action-irrelevant variables that neither change with actions nor influence actions' results, identified through conditional independence tests based on conditional mutual information. Their approach includes variables that are potentially useful for future tasks rather than being restricted to a particular training task.

### 4.1.2 Directed Exploration for Sample Efficiency

While a good representation of the environment is beneficial, it is not necessarily sufficient for sample efficient RL (Du et al., 2020). To improve sample efficiency, researchers have been studying directed exploration, strategies that guide the agent to explore specific parts of the state space that are believed to be more informative or more likely to yield high rewards. This can be done by giving bonuses to exploratory behaviors that discover novel or uncertain states. From a causal perspective, not all regions of high uncertainty are equally important. Only those that form a causal relationship with the success of the task are worth exploring. As an example, Seitzer et al. (2021) studied the problem of directed exploration in robotic manipulation tasks, where it is crucial for the agent to touch the target object to generate valuable data for learning complex manipulation skills. To this end, the authors proposed a method of measuring the causal influence of action on the object and incorporating it into exploration, greatly improving the sample efficiency of robotic manipulation tasks. On the other hand, Sontakke et al. (2021) introduced a method to learn self-supervised experiments based on the principle of independent causal mechanisms. This method leverages the idea that good experimental behavior should examine one causal variable at a time, while keeping other factors constant, resulting in a trajectory with relatively little information. Thus, the learning problem of experimental behavior is transformed into minimizing the amount of information contained in the generated data (also referred to as maximizing "causal curiosity"). Empirical results showed that RL agents pre-trained with causal curiosity can learn to solve new tasks more efficiently.

### 4.1.3 Data Augmentation for Sample Efficiency

Data augmentation is a common machine learning technique that aims to improve the performance of algorithms by generating additional training data. Counterfactual data augmentation is a causality-based approach that uses a causal model to imitate the environment and generate data that is not observed in the real world. This is particularly useful for RL problems because collecting large amounts of real-world data

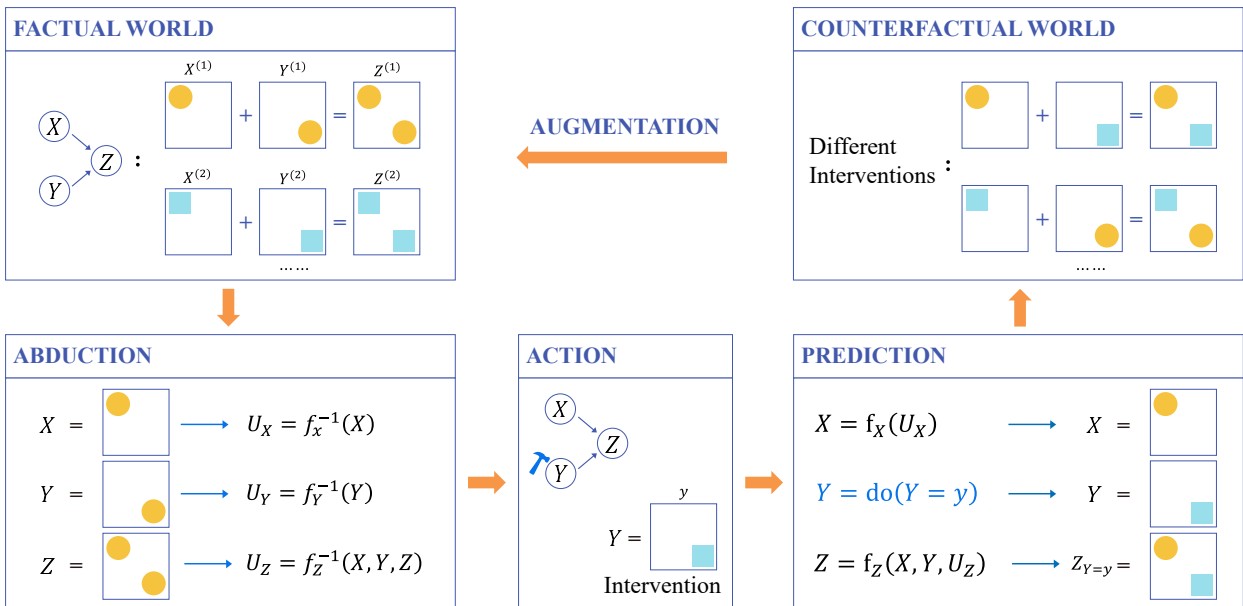

Figure 8: An example of counterfactual data augmentation following the counterfactual reasoning procedure: abduction, action, and prediction. The outcome of this procedure is then used to augment the training data observed in the factual world.

is often difficult or expensive. By simulating different counterfactual scenarios, RL agents can determine the effects of different actions without interacting with the environment, leading to more sample-efficient learning.

The implementation of counterfactual data augmentation follows a counterfactual reasoning procedure that consists of three steps (Pearl, 2009a), as demonstrated in Figure 8:

1. **Abduction** is about using observed data to infer the values of the exogenous variables $\mathcal{U}$;

2. **Action** involves modifying the structural equations of the variables of interest in the SCM; and

3. **Prediction** uses the modified SCM to generate counterfactual data by plugging the exogenous variables back into the equations for computation.

While MBRL methods can also generate samples with the learned models, they lack the means to model exogenous variables. This can result in underfitting when the distribution of exogenous variables become complicated (Buesing et al., 2019). In contrast, counterfactual data augmentation explicitly considers exogenous variables using the SCM framework and is able to generate higher-quality samples. From a Bayesian perspective, traditional MBRL approaches use a fixed prior distribution for exogenous variables, whereas counterfactual data augmentation uses observed data to estimate the posterior distribution.

Buesing et al. (2019) proposed counterfactually-guided policy search (CF-GPS), an algorithm for learning policies in POMDPs that uses SCMs to evaluate counterfactual actions. The CF-GPS method improves traditional model-based RL algorithms by inferring the posterior distribution of exogenous variables and considering alternative outcomes. Lu et al. (2020) proposed a sample-efficient RL algorithm to address the problems of mechanism heterogeneity and data scarcity. SCM is used to model the environment, which allows for counterfactual reasoning, enabling the agent to evaluate the potential consequences of alternative actions, thus avoiding actual exploration and mitigating bias due to limited experience. Pitis et al. (2020) presented local causal models (LCMs), derived from a global causal model by conditioning on a subset of state variables, to simplify counterfactual reasoning and increase its efficiency. Their proposed algorithm utilizes LCMs to generate counterfactual experiences and experimental results indicated significant improvements

regarding sample efficiency in various RL settings. More recently, Zhu et al. (2021) studied the application of counterfactual reasoning in robotic manipulation tasks, which increases the diversity of generated rollouts. The proposed method uses SCM to model the underlying dynamics and can generate counterfactual episodes with rarely seen or unseen objects to improve sample efficiency and generalization ability.

## 4.2 Causal Reinforcement Learning for Addressing Generalizability

Decision-making problems in the real world are ever-changing and hard to predict. Therefore, RL algorithms must be able to perform well in new and unseen situations during deployment, which is commonly referred to as generalizability. Generalization involves many types of problems. Zero-shot generalization requires the agent to only learn in training environments and be tested on the unseen. While this setting is appealing, it is sometimes infeasible in practice. Alternatively, adaptation (Zhang et al., 2015; Gong et al., 2016) assumes the agent can receive additional training in test domains, which encompasses a variety of settings such as transfer RL (Zhu et al., 2020), multitask RL (Vithayathil Varghese & Mahmoud, 2020), or lifelong RL (Khetarpal et al., 2022). A great deal of research has considered generalization in RL (Kirk et al., 2022), but there is still a lack of understanding of what capabilities an agent needs to achieve generalization and what generalization can be expected from a learning algorithm. Causal models offer a potential solution to these questions by disentangling factors that appear in the data generation process. This section classifies existing causal RL algorithms for generalization based on specific factors of change. The representative works are shown in Table 2.

### 4.2.1 Generalize to Different Environments

First, we consider how to generalize to different environments. From a causal perspective, different environments share most of the causal mechanisms but differ in certain modules, resulting from different interventions in the state variables. Based on the causal relationship between these variables, we can further divide existing works into two categories: generalizing to irrelevant variables and generalizing to different dynamics.

To enhance the ability to generalize to irrelevant factors, RL agents must examine the causality to identify the invariance in the data generation process. Zhang et al. (2020a) investigated the problem of generalizing to diverse observation spaces within the block MDP framework, which is a common scenario in reality, such as robots equipped with different types of cameras and sensors. In the block MDP framework, the observation space may be infinite, but we can uniquely determine the state (finite but unobservable) given the observation. The authors proposed using invariant prediction to learn the causal representation that generalizes to novel observations in the multi-environment setting. Similarly, Bica et al. (2021b) introduced invariant causal imitation learning, which learns the imitation policy based on invariant causal representation across multiple environments. Wang et al. (2022) studied the causal dynamics learning problem, which attempts to eliminate irrelevant variables and unnecessary dependencies so that policy learning will not be affected by these nuisance factors. Saengkyongam et al. (2022) studied the environmental shift problem under the framework of offline contextual bandits from a causal perspective. They proposed to find invariant policies based on an off-policy invariance test, a method that helps determine whether a subset of features remains invariant across different environments. The results suggested that invariance is key to obtaining distributionally robust policies when unobserved confounders are present. Ding et al. (2022) proposed a novel solution to the generalization problem in goal-conditioned reinforcement learning (GCRL) by treating the causal graph as a latent variable and optimizing it using a variational likelihood maximization approach. This method trains agents to discover causal relationships and learn a causality-aware policy, that is robust against changes in irrelevant variables.

Generalizing to new dynamics is a broader issue. It may involve variations in physical properties (e.g., gravitational acceleration, as shown in Figure 6d), differences between the simulation environment and reality, changes in the range of attribute values, etc. Sontakke et al. (2021) proposed training RL agents to infer and categorize causal factors in the environment with experimental behavior learned in a self-supervised manner.

---

[5]The term "toy" refers to simple, synthetically constructed datasets or simulation environments that are used to experimentally verify findings. It is not a concrete environment or task. We use this term consistently throughout the paper.

Table 2: Selected methods utilizing causality to improve generalizability.

| Category | Paper | Technique | Environments or Tasks | Source |
|---|---|---|---|---|
| Irrelevant variables | Zhang et al. (2020a) | Causal representation learning | Toy [5] | ICML |
| | | | Cart-pole (dm_control) | |
| | | | Cheetah (dm_control) | |
| | Bica et al. (2021b) | Causal representation learning | OpenAI Gym | NeurIPS |
| | | | MIMIC III | |
| | Wang et al. (2022) | Causal dynamics learning | Chemical | ICML |
| | | | Manipulation (robosuite) | |
| | Saengkyongam et al. (2022) | Causal representation learning | Toy | ICML Workshop |
| | Ding et al. (2022) | Causal discovery | Manipulation (Not accessible) | NeurIPS |
| | | | Unlock (Minigrid) | |
| | | | Crash (highway-env) | |
| Dynamics | Sontakke et al. (2021) | Causal representation learning | Manipulation (CausalWolrd) | ICML |
| | Lee et al. (2021) | Intervention | Manipulation (Isaac Gym) | ICRA |
| | | Domain randomization | | |
| | Zhu et al. (2021) | Counterfactual reasoning | Manipulation (CausalWolrd) | OpenReview |
| | Guo et al. (2022) | Mediation analysis | Pendulum (OpenAI Gym) | ICLR |
| | | | Locomotion (OpenAI Gym) | |
| Tasks | Eghbal-zadeh et al. (2021) | Causal representation learning | Contextual-Gridworld | ICLR Workshop |
| | Pitis et al. (2022) | Counterfactual reasoning | Spriteworld | ICML Workshop |
| | | | Pong (Roboschool) | |
| | | | Manipulation (OpenAI) | |
| Mixed | Zhang & Bareinboim (2017) | Causal reasoning | Toy | IJCAI |
| | Dasgupta et al. (2018) | Meta learning | Toy | ICLR Workshop |
| | Nair et al. (2019) | Causal induction | Light | arXiv |
| | Huang et al. (2022a) | Causal dynamics learning | Cart Pole (OpenAI Gym) | ICLR |
| | | | Pong (OpenAI Gym) | |
| Others | Zhu et al. (2022b) | Causal discovery | Toy | arXiv |
| | | Causal dynamics learning | Inverted Pendulum (OpenAI Gym) | |

These behaviors can help the agent to extract discrete causal representations from collected trajectories, which can be applicable to unseen environments, empowering the agent to effectively generalize to unseen contexts. Lee et al. (2021) explored the use of interventions to identify relevant state variables for successful robotic manipulation. The robot exhibited excellent sim-to-real generalizability after training with domain randomization on the relevant features. Zhu et al. (2021) developed an algorithm to improve the ability of agents to generalize to rarely seen or unseen object properties. This algorithm uses SCMs to model the environmental dynamics, allowing the agent to reason about what would have happened if the object had a different attribute value, which leads to improved generalizability. Guo et al. (2022) investigated the unsupervised dynamics generalization problem, which allows the model to generalize to new environments. The authors followed the intuition that data from the same trajectory/similar environments should have similar properties (hidden variables) that lead to similar causal effects. They used conditional direct effects in mediation analysis to measure similarity. The experimental results show that the learned model performs well in new dynamics.

### 4.2.2 Generalize to Different Tasks

Another important topic is how to generalize to different tasks. In the SCM framework, different tasks are created by altering the structural equation of the reward variable or its parent nodes on the causal graph. These tasks have the same underlying environmental dynamics, but the rewards are assigned differently.

Eghbal-zadeh et al. (2021) introduced causal contextual RL, in which the agent should learn adaptive policies that can adapt to new tasks specified by context variables. They proposed a contextual attention module that allows agents to incorporate disentangled features as contextual factors, achieving better generalization than non-causal agents. In order to make RL more effective in complex, multi-object environments, Pitis et al. (2022) suggested recognizing and utilizing local factors in transition dynamics, introducing a new framework named model-based counterfactual data augmentation, which generates counterfactual transitions based on the local structure, enabling the model to generalize to out-of-distribution tasks.

Furthermore, in reality, generalization may involve changes in both the environmental dynamics and the task. Several studies have explored this problem from a causal viewpoint. Zhang & Bareinboim (2017) investigated knowledge transfer across bandit agents in scenarios where causal effects are unidentifiable. The proposed strategy combines two steps: deriving causal bounds over the arms distribution based on structural knowledge and incorporating these bounds in a dynamic allocation procedure to guide the search towards more promising actions. The results indicated that this strategy dominates previously known algorithms and achieves faster convergence rates. Dasgupta et al. (2018) explored whether the ability to perform causal reasoning emerges from meta-learning on a simple domain with five variables. The experimental results suggested that the agents demonstrated the ability to conduct interventions and make sophisticated counterfactual predictions. Moreover, these emergent abilities can effectively generalize to new causal structures. Nair et al. (2019) studied the causal induction problem with visual observation. They incorporated attention mechanisms into the agent to generate a causal graph based on visual observations and use it to make informed decisions. The experiments demonstrated that the agent effectively generalizes to new tasks and environments with unknown causal structures. More recently, Huang et al. (2022c) proposed AdaRL, a framework for adaptive RL that learns a latent representation with domain-shared and domain-specific components across source domains. The latent representation is then used to learn an optimal policy parametrized by the domain-specific parameters. This framework allows for efficient policy adaptation to new environments, tasks, or observations, by estimating new domain-specific parameters with only a few samples.

### 4.2.3 Other Generalization Problems

In offline RL, the agent can only learn from pre-collected datasets. In this setting, agents may encounter previously unseen state-action pairs during the testing phase, leading to the distributional shift issue (Levine et al., 2020). Most existing approaches mitigate this issue through conservative or pessimistic learning (Fujimoto et al., 2019; Kumar et al., 2020; Yang et al., 2021b), rarely considering generalization to new states. Zhu et al. (2022b) proposed a solution to generalize to unseen states. They recovered the causal structure from offline data by using causal discovery techniques. The experimental results suggested that the causal world model exhibits better generalization performance than a traditional world model, and effectively facilitates offline policy learning.

### 4.3 Causal Reinforcement Learning for Addressing Spurious Correlations

As we stated in section 3.3, an RL agent is vulnerable to spurious correlations during the process of understanding the environment and task. Depending on the causal structure behind the decision-making problem, the spurious correlation can be one of two types: one corresponding to confounding bias caused by the fork structure and the other corresponding to selective bias caused by the collider structure ( Figure 7). We divide the existing methods into two categories accordingly. In particular, we also include work on imitation learning (IL) and off-policy evaluation (OPE), as they are both closely related to policy learning in RL. The representative works are shown in Table 3.

### 4.3.1 Addressing Confounding Bias

We start by introducing an important technique in causal inference named do-calculus (Pearl, 1995). It is an axiomatic system that enables us to replace probability formulas containing the do operator with ordinary conditional probabilities. The do-calculus includes three axiom schemas that provide graphical criteria for making certain substitutions. It has been proven to be complete for identifying causal effects (Huang & Valtorta, 2006; Shpitser & Pearl, 2006). Derived from the do-calculus, the backdoor and frontdoor adjustment are two widely used methods for eliminating confounding bias Pearl (2009b); Glymour et al. (2016). The key intuition is to block spurious correlations between the treatment and outcome variables that pass through the confounders.

The MAB problem can be thought of as a single-step decision-making problem with no state transitions. Forney et al. (2017) investigated a variant of this problem that involves unobserved confounders, which affect

---

[6]Causal graph as a technique refers to using causal graphs to describe the data generation process, and designing graphical criteria for determining properties such as identifiability or developing algorithms based on causal graphs.

Table 3: Selected methods utilizing causality to address spurious correlations.

| Category | Paper | Technique | Environments or Tasks | Source |
|---|---|---|---|---|
| Confounding bias - MAB | Forney et al. (2017) | Counterfactual data fusion | Toy | ICML |
| Confounding bias - MDP: IL | Zhang et al. (2020b) | Causal graph [6] | Toy | NeurIPS |
| | Kumor et al. (2021) | Causal graph | Toy | NeurIPS |
| | Swamy et al. (2022) | Instrumental variable regression | Lunar Lander (OpenAI Gym) Locomotion (PyBullet Gym) | ICML |
| Confounding bias - MDP: OPE | Namkoong et al. (2020) | Sensitivity analysis | Toy | NeurIPS |
| | Bennett et al. (2021) | Proximal causal inference | Toy | AISTATS |
| Confounding bias - MDP/POMDP | Lu & Lobato (2018) | Backdoor adjustment | Pendulum (OpenAI Gym) Cart Pole (OpenAI Gym) MNIST | arXiv |
| | Zhang & Bareinboim (2019) | Sensitivity analysis | Toy | NeurIPS |
| | Rezende et al. (2020) | Backdoor adjustment | Toy MiniPacman 3D Maze (Unity) | arXiv |
| | Zhang (2020) | Causal graph | Toy | ICML |
| | Wang et al. (2021b) | frontdoor adjustment Backdoor adjustment | - | NeurIPS |
| | Liao et al. (2021) | Instrumental variable regression | - | arXiv |
| | Gasse et al. (2021) | Do-calculus | Toy | arXiv |
| | Yang et al. (2022a) | Causal graph | Toy Cart Pole (OpenAI Gym) Lunar Lander (OpenAI Gym) | AAAI |
| | Zhang & Bareinboim (2022b) | Causal graph | Toy | NeurIPS |
| Selection bias | Bai et al. (2021) | Inverse probability weighting | Manipulation (OpenAI) | TCYB |
| | Deng et al. (2021) | Causal graph | D4RL | arXiv |

both actions and rewards. The authors first formally presented the regret decision criterion for maximizing the expected counterfactual reward conditioned on the intended arm. They then proved the counterfactual quantity can be estimated empirically, allowing for an informative combination of observational and interventional data for efficient learning without being affected by confounding bias. Zhang et al. (2020b) studied single-step imitation learning using a combination of demonstration data and structural knowledge about the data-generating process. They proposed a graphical criterion for determining the feasibility of imitation in the presence of unobserved confounders and a practical procedure for estimating valid imitating policies with confounded expert data. This approach was then extended to the sequential setting in a subsequent paper (Kumor et al., 2021). Swamy et al. (2022) designed an algorithm for imitation learning with corrupted data. They proposed to use instrumental variable regression (Stock & Trebbi, 2003), a well-known causal inference technique, to remove spurious correlations.

Several research focused on the off-policy evaluation (OPE) problem, which tries to estimate the performance of policies before they are deployed. For example, Namkoong et al. (2020) conducted sensitivity analyses on OPE methods under unobserved confounding. They developed worst-case bounds on the performance of an evaluation policy and proposed an efficient procedure for estimating these bounds with statistical consistency, allowing for a reliable selection of policies. Bennett et al. (2021), on the other hand, proposed a new estimator for OPE in infinite-horizon RL. The authors established the identifiability of the policy value from off-policy data by employing a latent variable model for states and actions (which can be seen as proxy variables for the unobserved confounders). The authors further presented an algorithm for estimating the stationary distribution ratio using proxies, which is then utilized to derive the policy value.

Lu & Lobato (2018) presented a method named deconfounding RL, which allows for learning effective policies from historical data affected by unobserved factors. This method utilizes backdoor adjustment to address confounding bias and outperforms traditional RL methods when applied to observational data with confounders. Liao et al. (2021) also focused on the offline setting. They found that unobserved confounders usually influence the actions in observational studies. They proposed an algorithm that helps efficiently identify the transition dynamics in RL using instrumental variables. Wang et al. (2021b), on the other hand, proposed a method for incorporating offline data in an online setting, accounting for confounding

variables that may affect the data's accuracy. This method effectively handles confounding bias by leveraging backdoor/frontdoor adjustment, and achieves a smaller regret than the optimal online-only approach. Gasse et al. (2021) studied a model-based method that combines interventional and observational data to learn a latent causal transition model, which is then used to solve the POMDP problem via deconfounding. Yang et al. (2022a) proposed an algorithm called causal inference Q-network to deal with the confounding bias raised by multiple types of observational interferences. The authors first analyzed how these interferences affect the decision-making process using causal graphs. They then designed a novel algorithm that learns to predict the interference label and adjust the policy accordingly. The experimental results suggested this algorithm is more resilient to observation interferences. Rezende et al. (2020) discussed the use of partial models in RL, a model-based approach that does not require modeling the full (and usually high-dimensional) observation. Partial models can be unreliable when they are confounded by the unmodeled aspects, leading to incorrect planning. To address this issue, the authors proposed a simple yet effective solution to ensure the causal correctness of partial models. They treated a node in the computational graph that is between the internal state and the action as a backdoor variable (e.g., the intended action), which allows for making causally correct predictions via backdoor adjustment.

A less familiar but highly important research topic for RL researchers is dynamic treatment regimes (DTRs) (Murphy, 2003). Closely related to the fields of biostatistics, epidemiology, and clinical medicine, this topic focuses on determining personalized treatment strategies, including dosing or treatment planning. It aims to maximize the long-term clinical outcome and can be modeled as an MDP with unobserved confounders (Wang et al., 2021b), making it highly relevant to causal RL. Zhang & Bareinboim (2019) considered a setting in which causal effect is not identifiable. They proposed an algorithm to solve DTRs by combining confounded observational data in online learning. Their algorithm hinges on sensitivity analyses and incorporates causal bounds to accelerate the learning process. In a follow-up work (Zhang, 2020), the authors focused on minimizing experimentation. They studied how to reduce the dimensionality of candidate policy space by exploiting the functional and independence restrictions encoded in the causal graph. Two novel online RL algorithms are developed to identify the optimal DTR, with one based on the principle of optimism in the face of uncertainty and the other built on posterior sampling. More recently, Zhang & Bareinboim (2022b) studied the DTR problem from a new perspective. They investigated the online RL of optimal policies with mixed scopes, i.e., finding the optimal combination of treatments, in an unknown SCM, provided with a causal graph encoding qualitative knowledge about the underlying model. The proposed method leveraged the causal graph to help evaluate the effects of a candidate policy from data collected by treatment regimes over different actions, achieving sublinear regret.

### 4.3.2 Addressing Selection Bias

Selective bias occurs when data samples fail to represent the target population. For example, selective bias arises when researchers seek to understand the effect of a certain drug on curing a disease by investigating patients in a selected hospital. This is because those patients may differ significantly from the population regarding where they reside, their social status, and their wealth, making them unrepresentative.

Bai et al. (2021) investigated the selective bias associated with using hindsight experience replay (HER) in goal-conditioned reinforcement learning (GCRL) problems. In particular, HER relabels the goal of each collected trajectory, allowing the agent to learn from failures (not reaching the original goal). The concern is that the relabeled goal distribution cannot properly represent the original goal distribution; thus, the agent trained with HER is biased. The authors proposed to use the inverse probability weighting technique in causal inference for learning, which allows the agent to improve sample efficiency with the help of HER while avoiding the bias that arises due from the relabeling, achieving promising results on a range of robotic manipulation tasks. Deng et al. (2021) viewed the offline RL problem through the lens of selective bias. An agent is vulnerable to the spurious correlation between uncertainty and decision-making in the offline setting, being prone to learning suboptimal policies. From the causal viewpoint, we can see that the empirical return is the outcome of both uncertainty and actual return. Since it is impossible to eliminate uncertainty by acquiring more data in the offline setting, a causal-unaware agent may mistakenly believe that there is a causal relationship between uncertainty and return. As a result, it prefers policies that achieve high returns by chance (high uncertainty). The authors propose quantifying uncertainty and using it as a penalty term

Table 4: Selected methods utilizing causality for goals beyond maximizing returns.

| Category | Paper | Technique | Environments or Tasks | Source |
|---|---|---|---|---|
| Explainability | Foerster et al. (2018) | Counterfactual | StarCraft | AAAI |
| | Madumal et al. (2020) | Counterfactual reasoning | OpenAI Gym
StarCraft | AAAI |
| | Bica et al. (2021a) | Counterfactual reasoning | Toy
MIMIC-III | ICLR |
| | Mesnard et al. (2021) | Counterfactual reasoning | Toy
Key-to-Door (Not accessible)
Interleaving (Not accessible) | ICML |
| | Tsirtsis et al. (2021) | Counterfactual reasoning | Toy
Therapy | NeurIPS |
| | Triantafyllou et al. (2022) | Counterfactual reasoning | Goofspiel (Not accessible) | AAAI |
| | Herlau & Larsen (2022) | Mediation analysis | Toy
DoorKey (Not accessible) | AAAI |
| Fairness | Zhang & Bareinboim (2018) | Counterfactual reasoning
Mediation analysis | Toy | AAAI |
| | Huang et al. (2022c) | Causal graph
Causal reasoning | Toy | AAAI |
| | Balakrishnan et al. (2022) | Causal graph
Counterfactual reasoning | Toy | AAAI |
| Safety | Hart & Knoll (2020) | Counterfactual reasoning | BARK-ML | IROC |
| | Everitt et al. (2021) | Causal graph | - | Synthese |

in the learning process. The results showed that this method outperforms various baselines that do not consider causality in the offline setting.

## 4.4 Beyond Return with Causal Reinforcement Learning

As automated decision systems based on RL become widely used in various activities of human society, there are growing concerns surrounding their usage. In this section, we explore ways to address or alleviate these concerns through a causal perspective. The representative works are shown in Table 4.

### 4.4.1 Explainable Reinforcement Learning via Causality

A good RL agent should not only effectively solve tasks but also provide clear explanations of its behavior. Large language models provide excellent examples of how causality can facilitate communication. Humans use causal language in everyday life, so a large language model trained on large-scale human-written text can communicate naturally and fluently with humans. In general, causal RL algorithms that model the data generation process using the SCM framework are inherently explainable. By providing prior knowledge (such as causal graphs) to the agents, we ensure that they share the same understanding of the environment with humans. In this way, the agent learns causality rather than correlations, allowing us to interpret the agent's decisions at the causal level rather than relying on ambiguous correlations.

In practice, we often desire granular explanations that involve counterfactuals. The term "counterfactual" is popular in multi-agent reinforcement learning (MARL). For example, Foerster et al. (2018) proposed a method named counterfactual multi-agent policy gradients for efficiently learning decentralized policies in cooperative multi-agent systems. More precisely, counterfactual solves the challenge of multi-agent credit assignment so that agents and humans can better understand the contribution of individual behavior to the team. Some subsequent studies followed the same idea (Su et al., 2020; Zhou et al., 2022). These approaches did not perform the complete counterfactual reasoning procedure as shown in Figure 8, missing the critical step of abduction, which offers opportunities for further enhancements. More recently, Triantafyllou et al. (2022) built a connection between Dec-POMDPs and SCM, allowing them to investigate the credit assignment problem in MARL using causal language. They proposed to formalize the notion of responsibility attribution based on the actual causality, as defined by counterfactuals, which is a significant stride in developing a

rigorous framework that supports accountable MARL research. Mesnard et al. (2021), on the other hand, studied the temporal credit assignment problem, i.e., measuring an action's influence on future rewards. Building on the notion of counterfactuals from causality theory, the authors suggested conditioning value functions on future events, which separate the influence of an agent's actions on future rewards from the effects of other random events. This method not only allows for credit assignment but also reduces the variance of policy gradient estimates.

Madumal et al. (2020) used theories from cognitive science to explain how humans understand the world through causal relationships and how these relationships can help us understand and explain the behavior of RL agents. They presented an approach that learns an SCM during reinforcement learning and used this model to generate explanations of behavior based on counterfactual analysis. They conducted a study with 120 participants. The results showed that the causality-based explanations performed better in understanding, explanation satisfaction, and trust than other explanation models. Bica et al. (2021a) discussed a method for understanding expert decision processes by integrating counterfactual reasoning into batch inverse RL. They proposed learning explanations of expert decisions by modeling their reward function in terms of preferences with respect to counterfactual outcomes, which is particularly helpful in real-world scenarios where active experimentation is often unfeasible. Tsirtsis et al. (2021) investigated how to find the optimal counterfactual explanation for a sequential decision process. They formalized this problem as a constrained search problem, i.e., how to search for another sequence of actions that differs from the observed sequence of actions by a specified number of actions. The work by Herlau & Larsen (2022) explores the application of mediation analysis in RL, whereby the agent is trained to optimize natural indirect effects, enabling it to identify critical decision points such as acquiring a key to open a door. By leveraging mediation analysis, the agent can learn a parsimonious and interpretable causal model, offering a novel approach to explainable RL.

### 4.4.2 Fair Reinforcement Learning via Causality

In addition to explainability, we also want RL agents to align with human values and avoid potential harm to human society, with fairness being a key consideration. Nevertheless, little work has been done to study fairness in RL. One popular idea is to quantify fairness using a statistical measure, such as demographic parity which requires an equal proportion of positive outcomes for each subgroup of a protected attribute (e.g. gender and age). This measure is then used to construct a constraint term (Balakrishnan et al., 2022) that enforces fairness, turning policy learning into a constrained optimization problem. However, fairness is commonly associated with a counterfactual problem: Would the results be different if the sensitive attribute had a different value? To accurately assess fairness, one must evaluate counterfactual quantities. Zhang & Bareinboim (2018) first introduced using the SCM framework to formulate the concept of fairness, which allows researchers to assess counterfactual fairness quantitatively. Using counterfactual statements, researchers can systematically analyze different types of discrimination and their effects on decision-making. Huang et al. (2022c) studied fairness in recommendation scenarios, focusing on the bandit setting (single-step decision-making), in which sensitive attributes should not causally influence the rewards. Liu et al. (2020) also investigated the fairness issue in the recommendation scenario. They studied how to balance accuracy and fairness in multi-step interactive recommendations modeled by MDPs. They utilized causal graphs to formally analyze fairness and evaluate it using counterfactuals. The experimental results showed that the proposed approach can enhance fairness while maintaining good recommendation quality.

### 4.4.3 Safe Reinforcement Learning via Causality

Finally, safety is a fundamental challenge to making RL widely applicable in the real world. Causal inference provides some valuable tools for studying safety. As an example, Hart & Knoll (2020) investigated the safety issue relates to autonomous driving. Researchers can conduct counterfactual policy evaluations before deploying any policy to the real world by utilizing counterfactual reasoning. The experimental results showed that their method demonstrated a high success rate while significantly reducing the collision rate. On the other hand, Everitt et al. (2021) studied the issue of reward tampering in sufficiently capable RL agents. These agents may find shortcuts to obtain rewards instead of performing the expected behavior, posing a

Table 5: The three components of causal learning.

|  | Available information | Targets to identify | Typical questions |
|---|---|---|---|
| **Causal feature extraction** | Observations | Causal variables (representation) | What factors account for the change in position? |
| **Causal discovery** | Causal variables | Causal graph | Does mass determine the change in position of an object? |
| **Causal mechanisms learning** | Causal graph | Causal mechanisms | How mass determine the change in position of an object? |

potential safety threat. The authors presented a precise and intuitive formalization of this problem using causal diagrams and proposed some useful design principles to prevent reward tampering.

### 4.5 Limitations

So far, we have highlighted the importance of causal RL, and reviewed the existing methods for addressing a range of research questions. However, causal RL also suffers from some fundamental challenges and limitations. As described in Definition 2.3, to incorporate causal information into the learning process, researchers must possess certain expert knowledge and make additional assumptions regarding the tasks and environments they aim to address. The correctness of these assumptions determines the effectiveness of causal RL, and incorrect assumptions may lead to adverse effects on RL. For example, if hidden confounders are not considered in the causal graph, the conclusions may be entirely opposite to reality (Pearl & Mackenzie, 2018). Moreover, it is a huge challenge to learn a causally correct model with limited prior knowledge. In certain scenarios, learning a causal model can be more challenging than directly learning policies, and may necessitate a large amount of interventional data, thus being counterproductive to sample efficiency gains. Although causal models provide additional advantages to RL, we must weigh their benefits and drawbacks. When dealing with counterfactual queries, the outcome of a different treatment cannot be observed by definition, just as we cannot ask a sailor suffering from scurvy to go back in time and intake a sufficient amount of vitamin C. Thus, counterfactual reasoning heavily relies on expert knowledge. Finally, having a perfect causal model does not solve all RL problems; how to use it effectively is the key. In the next section, we will further discuss some of the open problems and future directions of causal RL.

## 5 Open Problems and Future Directions

In this section, we consider some significant yet underexplored topics of causal RL.

### 5.1 Causal Learning in Reinforcement Learning

In the previous section, we explained how causality dynamics learning - a class of methods closely related to MBRL - can improve sample efficiency and generalizability (Wang et al., 2022; Huang et al., 2022b). These methods focus on understanding the cause-and-effect relationships between variables and the process that generates these variables. Instead of using complex, redundant connections, to model the data generation process, these methods prefer a sparse, modular style. As a result, they are more efficient and stable than traditional model-based methods and allow RL agents to adapt quickly to unseen environments or tasks. However, we may not have perfect knowledge of the causal variables a priori in reality. Sometimes, we must deal with high-dimensional and unstructured data like visual information. In this case, RL agents need to be able to extract causal representations from raw data (Schölkopf et al., 2021). Depending on the task, causal representations can be abstract concepts such as emotions and preferences, or they can be more concrete things such as physical objects.

The complete process of learning a causal model from data is known as causal learning (Peters et al., 2017). It is different from causal reasoning (Imbens & Rubin, 2015; Glymour et al., 2016), which only focuses on estimating specific causal effects given the causal model. Causal learning involves extracting causal features, discovering causality, and learning causal mechanisms. Table 5 briefly summarizes their characteristics. All three of these components are significant and deserve further investigation. A great deal of research has

been done on causal discovery (Spirtes et al., 2000; Pearl, 2009b; Peters et al., 2017; Vowels et al., 2022), a process of recovering the causal structure of a set of variables from data, particularly concerning conditional independence tests (Spirtes et al., 2000; Sun et al., 2007; Hoyer et al., 2008; Zhang et al., 2011). Under certain assumptions, such as faithfulness, algorithms can identify the Markov equivalence class of the underlying causal graph from observational data. Combining causal discovery with RL allows an agent to actively gather interventional data from the environment in an interactive way. Therefore, an interesting line of research in this field focuses on how to use interventional data or a combination of observational and interventional data for efficient causal discovery (Addanki et al., 2020; Jaber et al., 2020; Brouillard et al., 2020; Zhu et al., 2022a).

As for causal feature extraction, also known as causal representation learning (Schölkopf et al., 2021; Wang & Jordan, 2022; Shen et al., 2022), one possible method is to use an autoencoder to learn latent variables from high-dimensional observations (Yang et al., 2021a; Eghbal-zadeh et al., 2021; Tran et al., 2022). These methods can approximatively recover causal representations and structures by virtue of carefully designed constraint terms. The whole process is analogous to embedding an SCM into the learner, implicitly binding causal discovery and causal mechanisms learning in one solution. When multiple environments or tasks are available, one may also obtain causal representations through mining invariance (Zhang et al., 2020a; Bica et al., 2021b; Saengkyongam et al., 2022) or clustering trajectories (Sontakke et al., 2021). However, it is still challenging to determine the number and granularity of causal variables required for policy learning, and the optimal causal representations often depend on the specific context and task. Overall, the development of causal learning methods suitable for decision-making problems is an underexplored area and has the potential to advance the RL community. Conversely, RL techniques may also contribute to the field of causal learning (Zhu et al., 2022a).

## 5.2 Causality-aware Multitask and Meta Reinforcement Learning

Multitask RL (Parisotto et al., 2015; Teh et al., 2017; D'Eramo et al., 2020; Vithayathil Varghese & Mahmoud, 2020) focuses on solving multiple decision tasks simultaneously, which is often encountered in robot manipulation. For instance, a robot may need to acquire various skills or achieve multiple goals. Meta-learning (Duan et al., 2016; Finn et al., 2017; Gupta et al., 2018; Xu et al., 2018), on the other hand, involves training on a task distribution to gain the ability to adapt quickly to a new task. Both of these approaches are crucial in the practical application of RL. Researchers have achieved impressive results without considering causality. A natural question arises: Is it still necessary to consider causality if we can train a high-capacity model with a diverse range of tasks? Different tasks are inherently different interventions in the data generation process (Schölkopf et al., 2021). Therefore, if we can involve sufficiently diverse interventions during the training phase, the resulting model may emerge with strong generalization ability to solve unseen tasks efficiently.

Recent research has provided evidence that large, pre-trained models perform well in tasks requiring few-shot learning or even zero-shot generalization ability (Brown et al., 2020; Wei et al., 2021). Thus, it is reasonable to assume that large decision models (Reed et al., 2022; Wen et al., 2022) may also possess this ability. Interestingly, Dasgupta et al. (2018) further demonstrated that the capability of causal inference might emerge from large-scale meta-RL. However, testing all potential interventions and their combinations in real-world situations is impractical. This is where causality comes in. Causal models allow for the explicit incorporation of prior knowledge, enabling the model to align its understanding of the task or environment with human cognition. In addition, we can divide the agent's knowledge into multiple independent and autonomous modules. A non-causal agent would have to re-learn all the modules, even for a slightly changed task, while a causality-aware agent would only need to adapt a few modules, exhibiting a stronger knowledge transfer ability. This may also contribute to lifelong (or continual) learning (Xie & Finn, 2022; Khetarpal et al., 2022), allowing for fast adaptation to new tasks that arise in sequence.

## 5.3 Human-in-the-loop Learning and Reinforcement Learning from Human Feedback

Human-in-the-loop learning (HiLL) (Mosqueira-Rey et al., 2022) is a form of machine learning in which humans actively participate in the development cycle of machine learning models or algorithms. This can

involve providing labels, preferences, or other types of feedback. When the data or task being learned is complex or requires high levels of cognition, HiLL often produces better results because humans can provide valuable insights or knowledge to the model that it may be difficult for the model to learn on its own (Zhang & Bareinboim, 2022a).

In the context of RL, HiLL refers to involving humans in the MDP to replace the reward function that provides feedback signals. This allows us to train RL agents with the help of human knowledge and values rather than struggling to define a sophisticated reward function (Zhang & Bareinboim, 2022a). This idea is closely related to RLHF (Reinforcement Learning from Human Feedback) (Christiano et al., 2017), a concept that has gained increasing attention recently in the training of large language models (Ziegler et al., 2020; Glaese et al., 2022; Ouyang et al., 2022), where human instructors provide rewards (or penalties) to a model to encourage (or discourage) certain behaviors. From a causal perspective, humans can provide machine learning models with a strong understanding of causality based on their knowledge of the world, which can help filter out behaviors that may lead to negative outcomes. However, it is important to note that humans and machines may have different observations or perceptions of the world, and non-causal-aware RL agents may be influenced by confounding variables (Gasse et al., 2021). In addition, we often need to consider the issue of limited budgets, as our goal is to provide meaningful feedback to RL agents at the lowest possible cost. Finally, in addition to scalar feedback, we may also provide more informative feedback to agents in the form of counterfactuals (Karalus & Lindner, 2022).

## 5.4 Theoretical Advances in Causal Reinforcement Learning

The vast majority of research in the field has focused on the MAB problem. Lattimore et al. (2016) first introduced the causal bandits problem, where the agent can observe variables other than the reward after performing an action. The agent is expected to use this information to infer the causal structure and identify the arm with the highest reward more efficiently. Sen et al. (2017a) later built upon this work by developing a gap-dependent regret bound. Yabe et al. (2018) expanded upon this setting by allowing the agent to intervene on multiple nodes. Lee & Bareinboim (2018) formally demonstrated that non-causal agents are suboptimal and investigated the impact of non-manipulable variables in subsequent work (Lee & Bareinboim, 2019). Lu et al. (2021) studied the problem of causal bandits with prior causal knowledge, while Nair et al. (2021) focused on the problem with budget constraints in different causal structures. More recently, Kroon et al. (2022) eliminated the need for a priori causal knowledge by introducing the concept of "separating sets" in causal discovery. Similarly, Bilodeau et al. (2022) derived the optimal regret when observing a d-separator. Additionally, some research has focused on controlling confounding variables in the bandits problem Bareinboim et al. (2015); Sen et al. (2017b).

The multi-step decision-making problem is more complex than the MAB problem as it involves state transitions. Some studies focused on the dynamic treatment regimes (Zhang & Bareinboim, 2019; Zhang, 2020), which can be modeled as an MDP with a global confounder variable. There have also been studies on confounded MDPs (Zhang & Bareinboim, 2016; Wang et al., 2021b), a more general problem that assumes the existence of unobserved confounders at each time step. Overall, RL empowered by causality can achieve better theoretical bounds than non-causal approaches. The causal bandits problem and its variants have received much attention within the research community, but the MDP problem has not been studied as extensively. Moreover, in addition to understanding how causal models enhance the regret bounds, some work studied the identifiability of causal effects (Zhang et al., 2020b; Lu et al., 2022) or the underlying causal structure (Huang et al., 2022a). How to bring together theoretical results and practical solutions to create more effective causal RL algorithms is an open issue that is worth further attention and investigation.

## 5.5 Benchmarking Causal Reinforcement Learning

In RL, we are typically interested in the efficiency and convergence of the algorithm. Atari 2600 Games and Mujoco locomotion tasks (Brockman et al., 2016) are commonly used as benchmarks for discrete and continuous control problems. There are also experimental environments that evaluate the generalizability and robustness of RL, such as Procgen (Cobbe et al., 2020). Some benchmarks focus on multitask learning, meta-learning, and curriculum learning for reinforcement learning, such as RLBench (James et al., 2019), Meta-

World (Yu et al., 2021), Alchemy (Wang et al., 2021a), and Causal-World Ahmed et al. (2022). Among them, Causal-World delivers a diverse set of robotic manipulation tasks with a shared attribute set and structure that require the robot to construct a goal shape using the provided building blocks. This benchmark provides interfaces to manually modify objects' attributes (e.g., size, mass, and color), so researchers can intervene concerning these attributes in order to generate a series of tasks with the same causal structure.

Since causal RL is not limited to a particular type of problem, evaluation metrics may vary depending on the specific mission. While existing experimental environments have provided good benchmarks for evaluating algorithms with various metrics, the data generation processes within these environments are often opaque, hidden within game simulators or physics engines. This lack of transparency makes it difficult for researchers to fully understand the causal mechanisms behind the problems they are attempting to solve, hindering the development of the field. Recently, Ke et al. proposed a new set of environments focusing on causal discovery in visual-based RL, allowing researchers to specify the causal graph and adjust its complexity. However, as we demonstrated in section 4, much of the existing research still relies on toy environments to test the effectiveness of algorithms. Developing a suitable benchmark for causal RL remains an open question. In addition to the previously mentioned properties, a good benchmark should consider the multiple factors comprehensively, as discussed in section 4.4.

### 5.6 Real-world Causal Reinforcement Learning

Additionally, we note that there are currently very few real-world applications of causal RL. To make it more widely applicable, we must carefully address the various challenges posed by reality. Dulac-Arnold et al. (2020; 2021) identified nine critical challenges that are holding back the use of RL in the real world: limited samples; unknown and significant delays; high-dimensional input; safety constraints; partial observability; multi-objective or unspecified reward functions; low latencies; offline learning; and the need for explainability.

We have discussed some of these issues in paper, many of which are related to ignorance of causality. For instance, the challenge of learning from limited samples corresponds to the sample efficiency issue discussed in section 4.1. Learning from high-dimensional inputs and multiple reward functions relates to the generalization problem outlined in section 4.2. Offline learning raises concerns about spurious correlations (section 4.3), and security and explainability are covered in section 4.4.

Although causal models offer promising solutions to these real-world challenges, current experimental environments often fall short of meeting the research needs. As discussed in section 5.5, popular benchmarks are often treated as black boxes, and researchers have limited access to and understanding of the causal mechanisms by which these black boxes generate data. This situation significantly hinders the development of this research area. Therefore, establishing benchmarks that fully account for real-world factors would be valuable work that would also help increase the practical use of causal RL in the real world.

## 6 Conclusion

In summary, causal RL is a promising method of solving complex decision-making problems under uncertainty. It is an understudied but significant research direction. By explicitly modeling the causal structure of the target problem, causal RL algorithms can learn optimal policies more efficiently and make more informed decisions. In this survey, we aimed to clarify the terminologies and concepts related to causal RL and to establish connections between existing work. We proposed a problem-oriented taxonomy and systematically discussed and analyzed the latest advances in causal RL, focusing on how they address the four critical challenges facing RL.

While there is still much work to be done in this field, the results to date are encouraging. They suggest that causal RL has the potential to significantly improve the performance of RL systems in a wide range of problems. Here, we summarize the key conclusions of this survey.

- Causal RL is an emerging branch of RL that emphasizes understanding and utilizing causality to make better decisions.

- Causal modeling can enhance sample efficiency (section 4.1) and generalization ability (section 4.2); however, there are also fundamental challenges and limitations to consider (section 4.5). With limited causal information, RL agents may need to learn about causal representation and environmental dynamics from raw data (section 5.1).

- In proper conditions, causal relationships are identifiable (section 5.4), i.e., RL agents can recover causal relationships from observed data. Additionally, multitask learning and meta-learning can facilitate causal learning (section 5.2); in turn, causality can improve the knowledge transfer ability to solve various tasks more effectively.

- Correlation does not imply causation. Spurious correlations can lead to a distorted understanding of the environment and task, resulting in suboptimal policies (section 4.3). Apart from relying on causal reasoning techniques, we can also utilize human understanding of causality to aid RL (section 5.3).

- In real-world applications, performance is not the only concern. Other factors, such as explainability, fairness, and security, must also be considered (section 4.4). Current benchmarks require greater transparency and a comprehensive, multi-faceted evaluation protocol for reinforcement learning (section 5.5), which has significant implications for advancing real-world applications of causal reinforcement learning (section 5.6).

We hope this survey will help establish connections between existing work in causal reinforcement learning, inspire further exploration and development, and provide a common ground and comprehensive resource for those looking to learn more about this exciting field.

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

# A A Brief Introduction to Environments and Tasks

In this section, we briefly introduce the environments and tasks mentioned in section 4.

## A.1 Autonomous Driving

**BARK-ML**: `https://github.com/bark-simulator/bark-ml`. BARK is an open-source behavior benchmarking environment. It covers a full variety of real-world, interactive human behaviors for traffic participants, including Highway, Merging, and Intersection (Unprotected Left Turn).

**Crash (highway-env)**: `https://github.com/eleurent/highway-env`. The highway-env project gathers a collection of environments for autonomous driving and tactical decision-making tasks, including Highway, Merge, Roundabout, Parking, Intersection, and Racetrack. Crash is a modified version by (Ding et al., 2022) that is not publicly available.

## A.2 Classical Control

**Cart-pole (dm_control)**: `https://github.com/svikramank/dm_control`. Cart-pole is an environment belonging to the DeepMind Control Suite. It involves swinging up and balancing an unactuated pole by applying forces to a cart at its base.

**Acrobot (OpenAI Gym)**: `https://www.gymlibrary.dev/environments/classic_control/acrobot/`. The Acrobot environment consists of two links connected linearly to form a chain, with one end of the chain fixed. The goal is to apply torques on the actuated joint to swing the free end of the linear chain above a given height while starting from the initial state of hanging downwards.

**Mountain Car (OpenAI Gym)**: `https://www.gymlibrary.dev/environments/classic_control/mountain_car/`. The Mountain Car MDP is a deterministic MDP that consists of a car placed stochastically at the bottom of a sinusoidal valley, with the only possible actions being the accelerations that can be applied to the car in either direction. The goal of the MDP is to strategically accelerate the car to reach the goal state on top of the right hill.

**Cart Pole (OpenAI Gym)**: `https://www.gymlibrary.dev/environments/classic_control/cart_pole/`. A pole is attached by an un-actuated joint to a cart, which moves along a frictionless track. The pendulum is placed upright on the cart and the goal is to balance the pole by applying forces in the left and right direction on the cart.

**Pendulum (OpenAI Gym)**: `https://www.gymlibrary.dev/environments/classic_control/pendulum/`. The system consists of a pendulum attached at one end to a fixed point, and the other end being free. The pendulum starts in a random position and the goal is to apply torque on the free end to swing it into an upright position, with its center of gravity right above the fixed point.

**Inverted Pendulum (OpenAI Gym)**: `https://www.gymlibrary.dev/environments/mujoco/inverted_pendulum/`. This environment involves a cart that can moved linearly, with a pole fixed on it at one end and having another end free. The cart can be pushed left or right, and the goal is to balance the pole on the top of the cart by applying forces on the cart.

## A.3 Game

**MiniPacman**: `https://github.com/higgsfield/Imagination-Augmented-Agents`. MiniPacman is played in a $15 \times 19$ grid-world. Characters, the ghosts and Pacman, move through a maze.

**Lunar Lander (OpenAI Gym)**: `https://www.gymlibrary.dev/environments/box2d/lunar_lander/`. This environment is a classic rocket trajectory optimization problem. The landing pad is always at coordinates $(0, 0)$. The coordinates are the first two numbers in the state vector. Landing outside of the landing pad is possible. Fuel is infinite, so an agent can learn to fly and then land on its first attempt.

**Bipedal Walker (OpenAI Gym)**: `https://www.gymlibrary.dev/environments/box2d/bipedal_walker/`. This is a simple 4-joint walker robot environment.

**Car Racing (OpenAI Gym)**: `https://www.gymlibrary.dev/environments/box2d/car_racing/`. Car Racing is a top-down racing environment. The generated track is random in every episode. Some indicators are shown at the bottom of the window along with the state RGB buffer. From left to right: true speed, four ABS sensors, steering wheel position, and gyroscope.

**Beam Rider (OpenAI Gym)**: `https://www.gymlibrary.dev/environments/atari/beam_rider/`. The agent controls a space-ship that travels forward at a constant speed. The agent can only steer it sideways between discrete positions. The goal is to destroy enemy ships, avoid their attacks and dodge space debris.

**Pong (OpenAI Gym)**: `https://www.gymlibrary.dev/environments/atari/pong/`. The agent controls the right paddle, competing against the left paddle controlled by the computer.

**Pong (Roboschool)**: `https://github.com/openai/roboschool`. Roboschool is an open-source software for robot simulation, which is now deprecated. Pong allows for multiplayer training.

**Sokoban**: `https://github.com/mpSchrader/gym-sokoban`. This game is a transportation puzzle, where the player has to push all boxes in the room on the storage locations/ targets. The possibility of making irreversible mistakes makes these puzzles so challenging especially for RL algorithms.

**SC2LE**: `https://github.com/deepmind/pysc2`. SC2LE is a RL environment based on the StarCraft II game. It is a multi-agent problem with multiple players interacting. This domain poses a grand challenge raising from the imperfect information, large action and state space, and delayed credit assignment.

**VizDoom**: `https://github.com/Farama-Foundation/ViZDoom`. ViZDoom is based on Doom, a 1993 game. It allows developing AI bots that play Doom using only visual information.

## A.4 Healthcare

**MIMIC-III**: `https://physionet.org/content/mimiciii/1.4/`. MIMIC-III is a large, freely-available database comprising deidentified health-related data associated with over forty thousand patients who stayed in critical care units of the Beth Israel Deaconess Medical Center between 2001 and 2012.

**Therapy**: `https://github.com/Networks-Learning/counterfactual-explanations-mdp/blob/main/data/therapy/README.md`. This dataset contains real data from cognitive behavioral therapy. The data were collected during a clinical trial with the patients' written consent A post-processed version of the data is available upon request from `Kristina.Fuhr@med.uni-tuebingen.de`.

## A.5 Robotics

### A.5.1 Locomotion

**Cheetah (dm_control)**: `https://github.com/svikramank/dm_control`. Cheetah is an environment belonging to the DeepMind Control Suite. It is a running planar bipedal robot.

**OpenAI Gym**: `https://www.gymlibrary.dev/environments/mujoco/`. These environments are built upon the MuJoCo (Multi-Joint dynamics with Contact) engine. The goal is to make the 3D robots move in the forward direction by applying torques on the hinges connecting the links of each leg and the torso.

**PyBullet Gym**: `https://github.com/benelot/pybullet-gym`. This is an open-source implementation of OpenAI Gym MuJoCo environments using the Bullet Physics (`https://github.com/bulletphysics/bullet3`).

**D4RL**: `https://github.com/Farama-Foundation/D4RL`. D4RL is an open-source benchmark for offline RL. It includes several OpenAI Gym benchmark tasks, such as the Hopper, HalfCheetah, and Walker environments.

### A.5.2 Manipulation

**CausalWorld**: `https://github.com/rr-learning/CausalWorld`. CausalWorld is an open-source simulation framework and benchmark for causal structure and transfer learning in a robotic manipulation environment where tasks range from rather simple to extremely hard. Tasks consist of constructing 3D shapes from a given set of blocks - inspired by how children learn to build complex structures.

**Isaac Gym**: `https://github.com/NVIDIA-Omniverse/IsaacGymEnvs`. Isaac Gym offers a high performance learning platform to train policies for wide variety of robotics tasks directly on GPU.

**OpenAI**: The origin version is developed by OpenAI, known as "Ingredients for robotics research" (`https://openai.com/research/ingredients-for-robotics-research`), and now is maintained by the Farama Foundation (`https://github.com/Farama-Foundation/Gymnasium-Robotics`). It contains eight simulated robotics environments.

**robosuite**: `https://github.com/ARISE-Initiative/robosuite`. robosuite is a simulation framework powered by the MuJoCo physics engine for robot learning. It contains seven robot models, eight gripper models, six controller modes, and nine standardized tasks. It also offers a modular design for building new environments with procedural generation.

### A.6 Navigation

**Unlock (Minigrid)**: `https://github.com/Farama-Foundation/MiniGrid`. The Minigrid library contains a collection of discrete grid-world environments to conduct research on Reinforcement Learning. Unlock is task designed by (Ding et al., 2022), which is not publicly available.

**Contextual-Gridworld**: `https://github.com/eghbalz/contextual-gridworld`. Agents are trained on a group of training contexts and are subsequently tested on two distinct sets of testing contexts within this environment. The objective is to assess the extent to which agents have grasped the causal variables from the training phase and can accurately deduce and extend to new (test) contexts.

**3D Maze (Unity)**: `https://github.com/Harsha-Musunuri/Shaping-Agent-Imagination`. This environment is built on the Unity3d game development engine. It contains an agent that can move around. The environment automatically changes to a new view after every episode.

**Spriteworld**: `https://github.com/deepmind/spriteworld`. Spriteworld is an environment that consists of a 2D arena with simple shapes that can be moved freely. The motivation was to provide as much flexibility for procedurally generating multi-object scenes while retaining as simple an interface as possible.

**Taxi**: `https://www.gymlibrary.dev/environments/toy_text/taxi/`. There are four designated locations in the grid world. When the episode starts, the taxi starts off at a random square and the passenger is at a random location. The taxi drives to the passenger's location, picks up the passenger, drives to the passenger's destination (another one of the four specified locations), and then drops off the passenger. Once the passenger is dropped off, the episode ends.

### A.7 Others

**Chemical**: `https://github.com/dido1998/CausalMBRL#chemistry-environment`. By allowing arbitrary causal graphs, this environment facilitates studying complex causal structures of the world. This is illustrated through simple chemical reactions, where changes in one element's state can cause changes in the state of another variable.

**Light**: `https://github.com/StanfordVL/causal_induction`. It consists of the light switch environment for studying visual causal induction, where $N$ switches control $N$ lights, under various causal structures. Includes common cause, common effect, and causal chain relationships.

**MNIST**: `http://yann.lecun.com/exdb/mnist/`. The MNIST dataset contains 70,000 images of handwritten digits.

