# OpenReview forum: "Causal Reinforcement Learning: A Survey"
_TMLR — Rejected by TMLR_

### Review · Reviewer_GZue · 2023-03-01

**Summary Of Contributions:**

This paper provides a comprehensive review of causal reinforcement learning (RL), a subfield of RL that seeks to improve existing algorithms by utilizing the data generation process in terms of structural causal models. The authors explain how causal modeling can address challenges in non-causal RL. Also, they review existing causal RL approaches based on their target problems and methodologies. They discuss and analyze the latest advances in causal RL, focusing on how they address the four critical challenges facing RL. Finally, they conclude that causal RL has the potential to significantly improve the performance of RL systems in a wide range of problems.

**Audience:**

No

**Broader Impact Concerns:**

I don't find any concerns on the ethical implications of the work.

**Claims And Evidence:**

No

**Requested Changes:**



# 1. Introduction
* [C1]: Some undefined words can raise confusion. For example, what is causal machine learning (on page 2)? What is causal RL (on page 3)? Please define the terms before using them.

* [C2]. I think that the way the paper motivates causal RL is unclear. In the third paragraph on page 2, if the state transition and reward assignments are causal so that traditional RL can identify the causal relations between actions and rewards, then why must we consider causal RL? Given that traditional RL intervenes with the action, there is zero possibility that spurious (= non-causal) correlations could happen. The only RL scenarios where spurious correlation can be problematic are the off-policy settings, counterparts for observational studies. Please provide a detailed example of a task only causal RL can tackle.

* [C3]. The sentence, "Furthermore, causal knowledge can be represented quantitatively…" in the last paragraph on page 3, can mislead readers because it describes the causal graphs and the SCM as comparable frameworks. This is false because the SCM induces the causal graph, not vice versa. That is, all causal-graph-based methods implicitly assume an underlying SCM's existence. In more detail, the SCM is "sound and complete" [see Pearl (2009) Theorem 7.3.4 and Theorem 7.3.6] in the sense that any data-generating processes for counterfactual variables are reduced to the SCM. Therefore, I think the correct way of describing the causal model is to introduce the SCM and then the causal graphical method as a special case.

* [C4]. I don't understand the meaning of the sentence "However, they are not well connected with each other" in the first paragraph on page 3.

* [C5]. Overall, I don't think the Introduction section effectively conveys the importance of causal RL compared to traditional RL. Specifically, (1) there are no examples of what causal RL means; (2) the section does not explain the differences between causal RL (defined in question 1) and traditional RL; and (3) there are no motivational examples provided where causal RL succeeds while traditional RL fails. The only information I can gather about causal reinforcement learning (RL) is that it is an RL equipped with causal graphs induced by an unknown structural causal model (SCM) that represents the data-generating processes for action, reward, and state triplets. I cannot learn why causal RL is better than traditional RL from this introduction section because of the missing parts.

# 2. Background
* [C1]. Please explicitly state the assumption that the data-generating process is Markovian, meaning that all exogenous variables are independent and there are no latent confounders between endogenous variables.

* [C2]. In the first paragraph on page 5, please explicitly define what the counterfactual variables mean in the rubric of SCMs. Without the explicit definition, it’s hard to tell the difference between Y and Y_{X=x} from this paper.

* [C3]. I think "causal factorization" is not a formal term. A formal term for such factorization is to say, "P and G are compatible" or "P is Markovian relative to G" [Definition 1.2.2. of Pearl 2009]. You can also say that "P follows ordered local Markov condition w.r.t. G". However, "causal factorization" is not a formal term used in the community.

* [C4]. Continued from C3. I think the sentence “The former is causal, and the latter is not” misleads because such factorization from Bayesian Network (causal graph G) doesn’t encode the causality itself—the “Causal Bayesian Network” [Section 1.3. Pearl 2009] does.

* [C5]. I think [Section 2.3. Causal Reinforcement Learning] is unorganized. Specifically, the first two paragraphs made me think that an MDP represented in a rubric of an SCM is a causal RL.

* [C6]. I think Definition 2.3 is not formal. I believe causal RL is a subset of RL equipped with a causal graph induced by an SCM that focuses on additional knowledge/assumptions encoded in the graph to make better decisions. Please reflect this view in the definition if you agree. With the current definition, causal RL is a method of improving traditional RL without paying any costs. However, there are no magic solutions. Causal RL must be equipped with an additional assumption for analyzing causality, and this additional information is the key to improving traditional RL. Therefore, it’s important to explicitly state that causal RL is equipped with a causal graph to compare it transparently with traditional RL and study how additional assumptions could help.

* [C7]. I think the definition of the CRL should be moved to the first section because the causal RL has been used without being defined.

# 3. Why Causality Is Important in Reinforcement Learning
* [C1]. Most of the “why” in this section should be replaced with “how.” For example, the term “Why” in “Why CAusal Modeling Helps Improve Sampling Efficiency?” needs to be replaced with “how.”

* [C2]. Section 3.1.2. is written very vaguely without providing any concrete idea. Even if the so-called ‘causal modeling’ in the section is key to improving sample efficiency, the paper doesn’t provide any examples/descriptions of what causal modeling means. I believe the author meant ‘causal graph’ when saying ‘causal modeling’ because causal RL is more sample-efficient because it’s equipped with a causal graph. With a causal graph, RL algorithms can be efficient because agents can identify a subset of variables to intervene (e.g., variables with causal paths to the reward variable) instead of intervening in all sets of variables. Please write explicitly what causal modeling means and how it can improve sample efficiency with working examples.

* [C3]. The second paragraph of Section 3.1.2 doesn’t convey any information on how causal modeling helps sample efficiency. Specifically, I don’t see what the authors want to convey from the last sentence of the second paragraph.

* [C4]. The third paragraph of Section 3.1.2. is about the benefits of model-based RL, not causal RL.

* [C5]. Overall, there is a lack of demonstrations/examples comparing causal RL versus traditional RL methods for Section 3.1. For example, in Section 3.1.2, what kind of causal modeling can help reduce dimensionality? Please construct working examples for causal RL and traditional RL, and discuss the challenges based on these working examples.

* [C6]. Section 3.2.2. doesn't provide any information on how/why causal modeling helps generalization. I don't think that the modularity of the Bayesian Network is the reason for helping generalization. For any distributions, $P(V_1,V_2,...,V_n) = P(V_1)P(V_2 | V_1) \cdots P(V_n \vert V_{n-1},\cdots,V_1)$ holds. Then, intervening on $V_i$ only changes the distribution for $P(V_i \vert V_{i-1},\cdots,V_1)$. I think the reason why causal modeling can help generalization is due to its power to encode 'transportability' [https://arxiv.org/pdf/1503.01603.pdf] in the model.

* [C7]. In Section 3.3.1, I don't think that all the RL problems can face spurious correlation. For example, in an online learning setting, RL agents *intervene* in actions; there must be no spurious correlation. I think only a subset of RL problems, such as off-policy learning, can suffer from spurious correlation. Please clarify a category of RL problems and clearly distinguish which categories can suffer from spurious correlation.

# 4. Existing Work Relating to Causal Reinforcement Learning
* [C1]. In Table 1, the column "Source" is uninformative because this information can be obtained from the paper citation.
* [C2]. Table 1,2,3 are not comprehensible because they only list the abbreviated names and don't provide detailed information about Enironemts or Tasks.
* [C3] Overall, the summaries of cited papers are weak. For example, I cannot even guess what the cited papers are about from sumaries like "Wang et al., (2022), on the other hand, studied task-independent state abstraction" or "Saengkyongam et al., (2022) connected causality, invariance, and contextual bandit". Also, some of the cited papers were misunderstood in this review paper. For example, Zhang & Bareinboim (2017) is about using observational data to improve the bandit agent by using the lower/upper bound of causal effects that can be constructed from observational data. The summary "Zhang & Bareinboim (2017) used causal inference to improve knowledge transfer in reinforcement learning. Their method addresses the issue of transferring knowledge between bandit agents when standard techniques fail ... " is not a perfect summary because it doesn't contain how this paper used causal modeling to improve the bandit agents. Please improve the summaries of cited papers focusing on which causal modeling techniques were used and how these usages improved traditional RL agents.


**Strengths And Weaknesses:**

**Strength**: Readers can benefit from large lists and summaries of cited literature.

**Weakness**
1. I think that this paper could be better organized. For instance, it is unclear why the authors separated Section 3 and Section 4 because this division makes Section 3 somewhat vague. Section 3 lacks concrete examples of how causal RL can overcome obstacles that traditional RL faces. Some examples are presented in Section 4 as a list of cited papers related to these obstacles. Therefore, I suggest merging Sections 3 and 4 so that the contents in Section 4 can answer questions raised in Section 3.

2. This paper would benefit from working examples and demonstrations of causal RL methods compared to traditional ones. Without such examples, all discussions become somewhat vague. For example, in Section 3, the authors claimed that causal modeling for reinforcement learning methods could be more sample-efficient. However, without concrete examples to illustrate causal modeling, the paper becomes challenging to understand, and such equivocal terms make the discussion vague."

Overall, I think this paper requires substantial reorganization to be comprehensible. Also, the paper doesn't contain sufficient information on why/how causal RL works better than traditional RL.

---

> ### Author Response · Authors · 2023-03-27
> **Overall Response to Reviewer GZue**
>
> Thank you for the detailed review and the extensive feedback! We are encouraged that you found our paper comprehensively reviews the field and readers can benefit from large lists and summaries of literature. We added substantial new content based on your suggestions. The major concerns are addressed as below.
>
> > Lacks concrete examples of how causal RL can overcome obstacles that traditional RL faces
>
> Thank you for asking for more concrete examples. We have provided new examples and references in Section 3, including a new figure (Figure 5) that illustrates the difference between non-causal and causal state abstraction. We have also clarified the difference between regular and causal MBRL methods. Sections 3 and 4 are kept separate because they serve different writing purposes. Section 3 aims to provide insights before diving into the details in Section 4 by distilling and summarizing ideas from existing literature. We have added supporting references to improve the clarity. We hope that our updated manuscript addresses your concerns.
>
> > Equivocal terms make the discussion vague.
>
> Thank you for your valuable feedback. We appreciate your suggestion to provide a clearer definition of causal modeling and related concepts earlier in the paper. In response, we have added a new paragraph to the end of Section 2.3 that discusses different forms of causal modeling based on prior knowledge and research purpose. Furthermore, we have made an effort to use more specific terminologies throughout the paper to reduce ambiguity. We hope that these revisions will address your concerns and enhance the clarity and coherence of our work.
>
> Below, we provide a detailed description of the improvements we have made. If you have any further suggestions for enhancing the paper, please let us know and we will incorporate them accordingly.

---

> > ### Author Response · Authors · 2023-03-27
> > **Detailed Response to Reviewer GZue (1/n)**
> >
> > **Introduction**
> > > Please provide a detailed example of a task only causal RL can tackle.
> >
> > We have added an example of scurvy in Section 1, which is used throughout the paper, to highlight the need for causal RL. Traditional RL methods often require massive exploratory behaviors, which can be costly and even fatal. In contrast, with causal RL, one would analyze the causal relationships and make wise assumptions beforehand, avoiding meaningless and superstitious attempts; thereby increasing efficiency and safety. Furthermore, causal inference helps to estimate causal effects, preventing us from drawing wrong conclusions.
> >
> > > "However, they are not well connected with each other".
> >
> > We note that there is a lack of clarity and coherence in the existing literature on causal RL, primarily because causal modeling is more of a mindset than a specific issue. This motivates us to write this survey paper.
> >
> > **Background**
> > > I think "causal factorization" is not a formal term.
> >
> > Thank you for your comments. The term 'causal factorization' was introduced in (Schölkopf et al., 2021). We have revised the manuscript according to your opinion and included relevant references.
> >
> > > “The former is causal, and the latter is not” misleads because such factorization from Bayesian Network (causal graph G) doesn’t encode the causality itself
> >
> > Thanks for the comments. We have clarified that the former decomposition conforms to the causal graph of the given example, and revised the context based on your suggestions.
> >
> > > Definition 2.3 is not formal. ...
> >
> > Thank you for your valuable comments. We agree that causal RL relies on additional assumptions to incorporate causal information into learning, and have revised the definition to reflect this emphasis.
> >
> > **Why Causality Is Important in Reinforcement Learning**
> > > Most of the “why” in this section should be replaced with “how.”
> >
> > Thank you for your comments. We have rewritten Section 3 in the updated manuscript. Specifically, we have adjusted the text to better compare the difference between non-causal and causal methods, and we have provided new examples and supporting references for the claims made in this section. We hope these changes better answer the "why" questions and improve the overall clarity of our manuscript.
> >
> > > I don't think that the modularity of the Bayesian Network is the reason for helping generalization.
> >
> > Thank you for your constructive comments. We have carefully considered your suggestions and revised the manuscript to better clarify the connection between causality and generalizability. Following the interpretation of transportability in [1] and [2], it is the invariance assumption that opens the door for the transport of experimental findings. We have added a discussion of the independent causal mechanism and sparse mechanism shift hypothesis (Schölkopf et al., 2021; Perry et al., 2022) in Section 3.2.2 and used a concrete example to illustrate why invariance helps generalization and knowledge transfer.
> >
> > > In an online learning setting, RL agents _intervene_ in actions; there must be no spurious correlation.
> >
> > Thank you for your comment. We want to clarify that we did not claim all RL problems face spurious correlations and respectfully disagree that online learning necessarily implies no spurious correlations. In Section 3.3.2, we provide a concrete example showing that spurious correlations can indeed occur in online RL. In personalized recommendation, if the clicked items always correlate with spurious features such as item popularity and clickbait titles in the training environment, an RL agent may learn a policy based on these features, leading to incorrect decisions in the test environment.
> >
> > [1] Pearl, Judea, and Elias Bareinboim. "External validity: From do-calculus to transportability across populations." _Probabilistic and causal inference: The works of Judea Pearl_. 2022. 451-482.
> >
> > [2] Mao, Chengzhi, et al. "Causal transportability for visual recognition." _Proceedings of the IEEE/CVF Conference on Computer Vision and Pattern Recognition_. 2022.

---

> > > ### Author Response · Authors · 2023-03-27
> > > **Detailed Response to Reviewer GZue (2/n)**
> > >
> > > **Existing Work Relating to Causal Reinforcement Learning**
> > > >  In Table 1, the column "Source" is uninformative because this information can be obtained from the paper citation.
> > >
> > > While the information can be obtained from the paper citation, we keep this column to provide a quick and easily accessible summary of the venues where the latest research on causal RL appears, saving readers time in checking the references.
> > >
> > > > Table 1,2,3 are not comprehensible because they only list the abbreviated names and don't provide detailed information about Enironemts or Tasks.
> > >
> > > Thank you for your constructive feedback. We have addressed this concern in the revised manuscript by adding a detailed description of the environments and tasks in Appendix A.
> > >
> > > > improve the summaries of cited papers
> > >
> > > Thank you for your valuable comments. We have taken them into account to carefully revise Section 4. Specifically, we have further explained the meaning of task-independent state abstraction in Wang et al. (2022), highlighted the use of invariance to solve the contextual shift problem in offline contextual bandits in Saengkyongam et al. (2022), and adjusted the wording in Zhang & Bareinboim (2017) to emphasize the use of causal bounds for knowledge transfer on bandit problems where causal effects are not identifiable. We ensure that each review clearly states which technique was used, what problem the paper solved using this technique, and provides necessary descriptions of any terminology used. We hope these improvements make Section 4 more informative and useful for readers.
> > >
> > > **Minor adjustments**
> > >
> > > Thank you for your thoughtful comments. We have made adjustments based on your suggestions.
> > > - "Some undefined words can raise confusion. ": We have removed the term "causal machine learning" and added an explaination of causal RL in Section 1. We keep the definition 2.3 in Section 2 for clarity and coherence.
> > > - "Please explicitly state the assumption that the data-generating process is Markovian.": We have added the assumption in Section 2.1.
> > > - "Please explicitly define what the counterfactual variables mean in the rubric of SCMs.": We have modified the text to make the definition of counterfactual variables clearer.
> > > - "Furthermore, causal knowledge can be represented quantitatively…": We have removed this statement and formally described the relationship between SCM and causal graph in Section 2.1.
> > > - "an MDP represented in a rubric of an SCM is a causal RL.": We have adjusted the content to better explain the transformation and emphasized that it does not constitute causal RL on its own.
> > > - "The third paragraph of Section 3.1.2. is about the benefits of model-based RL, not causal RL.": We have revised the discussion to better explain the difference between non-causal and causal MBRL.

---

### Review · Reviewer_swrH · 2023-03-04

**Summary Of Contributions:**

This paper gives a comprehensive review of causal reinforcement learning, including answering what causal RL is, why it needs to be studied, and how causal modeling improves existing RL approaches. It also includes a clear overview of the foundational
concepts of causal RL research and identifies the bottleneck problems in RL that can be solved or improved by means of causal modeling.

**Audience:**

Yes

**Claims And Evidence:**

Yes

**Requested Changes:**

No

**Strengths And Weaknesses:**

I am delighted with this review paper. It is very comprehensive and organized very well. To my knowledge, it is the first comprehensive survey of causal RL.

I am happy with the current version.

---

> ### Author Response · Authors · 2023-03-27
> **Response to Reviewer swrH**
>
> Thank you very much for your positive feedback regarding our manuscript!

---

### Review · Reviewer_B1jj · 2023-03-17

**Summary Of Contributions:**

This paper is presented as a survey of the field of "Causal Reinforcement Learning". It consists of four parts:
 - Section 2 presents a brief introduction to reinforcement learning (RL) and Judea Pearl's SCM framework of causality, which is then used to propose a common framework for causal reinforcement learning
 - Section 3 presents arguments as to why causal modeling can improve sample efficiency, improve generalization, facilitate knowledge transfer, address spurious correlation, achieve explainability, achieve fairness, and achieve safety in RL
 - Section 4 presents existing works, grouped into categories (sample efficiency, generalizability, spurious correlation, explainability, fairness, safety)
 - Section 5 presents open problems

**Audience:**

Yes

**Broader Impact Concerns:**

No concern

**Claims And Evidence:**

No

**Requested Changes:**

The paper must make it clearer in its claims about causal RL what are opinions from the authors, and what are facts substantiated by evidence (in particular, all the answers to the questions in Section 3). I would be much more happy with the authors presenting these questions as open, then reviewing papers that attempt to demonstrate positive or negative answers, and finally summarizing with a more evidence-based and nuanced answer (e.g., pinpoint the limitations and the remaining challenges). This would I believe be much more beneficial to practitioners and researchers in the field, as well to newcomers. This point is critical to me. At the moment I do not find the paper claims to be well-supported by evidence, which is a major criterion for TMLR.

The authors must make it clearer what the concepts of causal modeling, causal learning, causal reasoning, causal representations, causal models, and causal RL refer to. Even if these are broad concepts (I like for example the broad definition 2.3), it is very confusing that these different notions are interchangeably used in the text.

The authors must make it clearer whether the presented SCM formulation of causal RL provides a universal framework to reason in the field, or rather if it is just an illustrative example of causal modeling. Given the wide range of topics and papers discussed in the survey, I find this particular formulation a bit limited, and maybe not very relevant. The introduction claims "Our survey of causal RL presents a comprehensive overview of the field, aligning existing research
within the SCM framework", but it seems that none of the papers discussed in section 4 (existing works) is discussed under the perspective of this SCM formulation.

**Strengths And Weaknesses:**


### Strengths

The paper compiles a broad range of works related to causality and RL, which I believe is relevant and might be useful to the community.

### Weaknesses

My main criticism is that I found the paper to not give a clear account of the challenges and limitations of causality for RL, and to carry a lot of ambiguity in its message. I completely agree with the authors that "there is a lack of clarity and coherence in the existing literature on this topic, primarily because causal modeling is more of a mindset than a specific issue". Unfortunately, I also find that this survey paper does not do a very good job at addressing this issue, and rather contributes to perpetuate it.

 - The paper, especially Section 3, presents causal modeling as the missing piece that can solve all the problems faced in RL (sample efficiency, generalization, spurious correlations, explainability, fairness, safety), without question. For example in 3.1.2. "Causal representations have better robustness and transferability than correlation-based representations", "Causal modeling helps the agent detect the key regions, such as the region near the target in robotic manipulation, by narrowing down the scope", "Causal models are more powerful than traditional correlation-based models because they are more robust and enable counterfactual reasoning". I do not find these claims very convincing, as they are given as plain assertions without any supportive evidence or reference. Are these universal truths? Are they the author's opinion? Do they summarize widespread hopes in the field? The entire section 3 is rather ambiguous in that respect, and even a bit dogmatic.

 - I have the same concern regarding the SCM formulation of causal RL provided in Section 2.3. It is unclear whether this proposed formulation is universal and encompasses all of the field of causal RL (which seems to be implied in the text), or if it is just an illustrative example. Again the text is very ambiguous in that regard. I do not believe that the proposed SCM formulation can provide a unified framework for the field, and I find sentences such as "The causal graph and the SCM that correspond to an MDP are shown in Figure 4" And "The SCM framework allows us to discuss causality in decision-making problems" to be misleading in this direction.

 - I praise the authors for this sentence in Section 4 "there is a lack of clarity and coherence in the existing literature on this topic, primarily because causal modeling is more of a mindset than a specific issue", which I find very much to the point for summarizing the field. But this is also I believe the first tentative description of what causal modeling is in the paper, while the term is used extensively in the previous section. This definition should be provided and discussed much earlier in the paper, in order to put things in perspective.

 - The paper dedicates 5 pages to the praise of causal modeling in Section 3, I think it would be fair to the readers to spend some space and effort to describe the fundamental challenges and limitations as well, such as the fact that causal learning (causal modeling?) requires either interventions or expert knowledge (layer 2 in Pearl's ladder), and that counterfactual reasoning can not be validated experimentally and must also rely on expert knowledge (layer 3 in Pearl's ladder). These challenges are barely touched only at the end of the paper in Section 5.1.

### Detailed comments (reading notes)

 - p2 Data alone cannot... -> This statement is a bit vague and misleading. Data collected from experience (experimental data, a.k.a. interventional data) can answer causal questions. I'd suggest to rephrase, something in the line of "Collecting data by observation". As this section sets the tone for the rest of the paper, I think it is important to be clear here.

 - p2 Data-driven [...] and taste -> These two sentences are misleading as well. The issue is not with statistical learning or machine learning algorithms, but with the type of data fed into these algorithms. Also, the term "data-driven machine learning" is a pleonasm. I do believe that all machine learning approaches are data-driven, by definition.

 - p2 RL agents equipped [...] policy evaluation -> How does this differ from regular model-based RL? This statement is misleading. It seems to imply that modeling the environment as an SCM is a necessary condition, in order to generate data without acting in the environment.

 - p2 This difficulty is ... -> Why would that be? An explanation is missing here.

 - p2 RL agents equipped [...] policy evaluation -> How does this differ from regular model-based RL? This statement is misleading. It seems to imply that modeling the environment as an SCM is a necessary condition, in order to generate data without acting in the environment.

 - p3: the central role of causality in human intelligence -> This is a bold statement. Human intelligence is not well understood to this date. A supporting reference at least would be nice here.

 - p3: extant literature -> typo?

 - p3: Structual -> typo

 - p3 Definition 2.1: Something missing in this definition is that F must induce an acyclic graph (DAG)

 - p4 Figure 2: The nodes corresponding
to the exogenous variables are usually omitted in the causal graph. -> Why adopting the SCM framework then? Why not simply relying on causal graphs?

 - p4: Structual -> typo

 - p4: is an independent variable in -> I am puzzled by this concept of independent variable in a structural equation. Is an independent variable an input variable? In which sense can such a variable be considered independent? Independent of what?

 - p4: all distributional shifts result [...] in the data generation process -> I do not agree with that statement. From the same data generation process, one can generate two distributions by selectively choosing samples from it based on arbitrary criteria. This is not an intervention, but a post-hoc re-sampling which result in a different distribution.

 - p4: "while making changes [...] exogenous variables)" -> How can the structural equations change the distribution of exogenous variables? According to Definition 2.1 the structural equations affect only the endogenous variables.

 - p5 Figure 3: It took me a while to parse the figure and understand the message. Which variable is what is not clear, in particular when compared with Figure 2. It seems here that Y is the variable "what did sailors eat", which can take three values (fresh citrus, cooked citrus, rotten meat). What is displayed in the images is not a distribution over Y, and it is not even a distribution over X. Also, in Figure 2 the image of a citrus seems to denote a random variable (like X), while here it seems to denote a value (like 1 in X=1, or X=2, X=3). This is all confusing even to me, a reader already familiar with do-calculus and counterfactual reasoning.

 - p5: In particular, [...] they overlap -> this is in general not true. Let me use (x,y,z) for variables in the real world, and (x',y',z') in the imaginary world. Clearly, in the SCM from Figure 5, $p(x'|do(y'),do(y),x)$ is not equal to $p(x|do(y))$.

 - p6: Causal factorization [...] causal graph structure -> I do not believe this statement to be true. Any reference to support that claim? The causal graph is not necessarily sparse. Also, what about situations where not all variables of the causal system are observed (hidden variables)? How does one exploit the causal factorization then? I'd suggest to either remove that sentence or substantiate it with evidence.

 - p6: causal factorization facilitates generalization to new problems -> This is vague and ambiguous. What is meant by a new problem here? A control problem? A pure prediction (statistical learning) problem? An intervention on a single variable indeed alters only the causal mechanism on that variable, but not all new problems arise from an intervention. You might want to transfer a model trained on a population A to a population B, or from a training dataset to a test dataset, or from a simulated environment to a real-world environment. These situations are not all the result of an intervention. Finally, even considering only situations where a "new problem" originates from an intervention, it seems that this general statement relies on the assumption that all random variables in the causal system that generated the data are observed. What about hidden variables? Consider image samples, what is the causal factorization? Does it always exist?

 - p6: Definition 2.2 seems to apply only to discrete action and state spaces. This should be clearly stated.

 - p7 Section 2.3: The content of this section is ambiguous. It is unclear if what is presented here is a universal causal formulation of RL, if it is restricted by arbitrary modeling choices from the authors, and what are the consequence of these choices. I do believe that the proposed formulation is limiting in terms of which counterfactuals can be expressed and computed based on these independent exogenous variables. This should at the very least be clarified.

 - p7: From a causal perspective [...] in the SCM -> This reformulation of an MDP as an SCM is arbitrary. Why would the (PO-)MDP causal graph not be sufficient to reason about causaility in RL? What does this (particular) SCM formulation bring to the table? Why would the exogenous variables be independent? Why would there be exogenous variables at all? Is it purely for convenience (the reparameterization trick) or is there something more profound about these choices? Do these choices imply limitations in terms of causal reasoning in RL (computing counterfactuals etc.)? A discussion is missing.

 - p7: the policy is not a causal relationship -> Why not? Again, this seems like an arbitrary choice, which is not clearly stated.

 - p8 Definition 2.3: I like this broad definition of causal RL. However, later on the term "Causal Modeling" is used (Sections 3.1.2, 3.2.2, 3.3.2). What is causal modeling? Does it involve learning the causal graph? Learning the causal mechanisms, given an a-priori graph? Both?

 - p8 Section 3: This section presents causal modeling as the missing piece that has the potential to solve all the problems faced in RL. But it does a poor job at explaining what the new challenges are. Say that all of these problems are solved if one possesses the true underlying causal model of the environment. Obtaining this causal model is not achieved simply by reformulating an MDP as an SCM, as suggested in Section 2.3. The paper dedicates 5 pages to the praise of causal modeling in Section 3, I think it would be fair to the readers to spend some space and effort to describe the fundamental challenges and limitations as well, such as the fact that causal learning (causal modeling?) requires either interventions or expert knowledge (layer 2 in Pearl's ladder), and that counterfactual reasoning can not be validated experimentally and must also rely on expert knowledge (layer 3 in Pearl's ladder). These challenges are barely touched only at the end of the paper in Section 5.1.

 - p8: a high training costs -> typo

 - p9 Section 3.1.2: It is hard to distinguish here what is literature-based evidence and what is opinion and speculation from the authors. This ambiguity should be clarified.

 - p10: In general, changes [...] -> How so? Why? Supporting evidence is missing.

 - p11 Figure 6: In the 3rd graph, Z is not a collider node.

 - p13: there is a lack of clarity and coherence in the existing literature on this topic, primarily because causal modeling is more of a mindset than a specific issue -> I find this sentence very much to the point. A discussion around this issue is missing in the paper. Section 2.3 would be a good place for such a discussion.

 - p17: the observation may be infinite [...] (finite but unobservable) -> This statement is ambiguous, I am not sure what is meant here.

 - p17: Saengkyongam et al. (2022). connected -> typo

 - p19 Table 3: The text says that Gasse et al. (2021) address the confounding problem in POMDPs, while the table indicates MDP

 - p19: one of the most common [...] do-calculus -> the front-door and back-door criterions are consequences of do-calculus, not alternate solutions. If the causal query is identifiable the do-calculus will appropriately return a front-door formula, a back-door formula, a combination of both, or something more involved.

 - p26: In summary, causal RL is a promising method of solving complex decision-making problems under uncertainty. It is an understudied but significant research direction. By explicitly modeling the causal structure of the target problem, causal RL algorithms can learn optimal policies more efficiently and make more informed decisions. -> Again, this conclusion appears rather dogmatic to me. It does not highlight the limitations of the existing works in causal RL, the fundamental challenges, and what remains to be done. Apart from providing a large compilation and categorization of the research papers in the field, I find the hindsights and the conclusions rather limited for a survey paper.

 - p26: Causal modeling can enhance [...] from raw data -> It would make much more sense to introduce first the challenge, learning a causal model from data, and then discuss the solutions proposed in the literature. As the paper is framed now, it is unclear how the papers listed in sections 4.1 and 4.2 address this challenge.

---

> ### Author Response · Authors · 2023-03-27
> **Overall Response to Reviewer B1jj**
>
> Thank you for the detailed review and the extensive feedback! We are encouraged that you found our paper compiles a broad range of works and is relevant and useful to the community. We added substantial new content based on your suggestions. The major concerns are addressed as below.
>
> > Clarify the sources of the claims in the pape
>
> Thank you for requesting clearer claims. We want to assure you that throughout the paper, we only make claims that are well-supported by evidence. In Section 3, we summarized the claims from the existing literature presented in Section 4 to provide insights before diving into the details. We recognize that this section could be clearer, and we have rewritten it in the updated manuscript. Specifically, we have added new examples and a new diagram (Figure 5), and we have provided supporting references for the claims made in this section. We hope these changes address your concerns and improve the overall clarity of our manuscript.
>
> > Describe the fundamental challenges and limitations
>
> Thank you for the useful comment. We have adjusted the tone of our manuscript to present the challenges in Section 3 as open. We have also revised the manuscript by adding a subsection (Section 4.5) that discusses the fundamental challenges and limitations of causal RL. Incorporating causal information into the learning process requires making additional assumptions, and incorrect assumptions can have adverse effects on RL. Moreover, learning a causally correct model with limited prior knowledge can be challenging, and may require a large amount of interventional data, which can be counterproductive to sample efficiency gains. Although causal models provide additional advantages to RL, we must weigh their benefits and drawbacks, and having a perfect causal model does not solve all RL problems. We hope that this discussion will provide readers with a balanced perspective on causal RL. Building on this discussion, Section 5 further discusses the open problems in causal RL and what efforts researchers have made to address them.
>
> > I praise the authors for this sentence in Section 4 "there is a lack of clarity and coherence in the existing literature on this topic ...", ... Clarify what the concepts of causal modeling, ..., refer to. Section 2.3 would be a good place for such a discussion.
>
> Thank you for your valuable feedback. We appreciate your suggestion to provide a clearer definition of causal modeling and related concepts earlier in the paper. In response, we have added a new paragraph to the end of Section 2.3 that discusses different forms of causal modeling based on prior knowledge and research purpose. Furthermore, we have made an effort to use more specific terminologies throughout the paper to reduce ambiguity. We hope that these revisions will address your concerns and enhance the clarity and coherence of our work.
>
> > Whether the presented SCM formulation of causal RL provides a universal framework to reason in the field. ... This reformulation of an MDP as an SCM is arbitrary.
>
> We appreciate the comment and have revised the manuscript to clarify that the figure in Section 2.3 is an illustrative example that can be adjusted to different levels of granularity based on prior knowledge. We want to emphasize that casting an MDP into an SCM does not introduce additional constraints and is always possible (Buesing et al., 2019). Our use of the SCM formulation is not meant to imply that all causal RL methods must learn an SCM. Instead, we aim to provide a flexible framework for describing causality in RL that can accommodate various forms of causal modeling.
>
> Below, we provide a detailed description of the improvements we have made. If you have any further suggestions for enhancing the paper, please let us know and we will incorporate them accordingly.

---

> > ### Author Response · Authors · 2023-03-27
> > **Detailed Response to Reviewer B1jj (1/n)**
> >
> > > Data-driven ... and taste -> The issue is ... with the type of data fed into these algorithms ... all machine learning approaches are data-driven ...
> >
> > Thanks for the comment. To clarify, this statement is based on the perspective presented by Pearl and Mackenzie (2018). We have added a footnote to the updated manuscript to clarify that "pure" data-driven methods refer to approaches that solely focus on mining or summarizing data, without considering underlying mechanisms. “the type of data fed into algorithms” implies we are actually making an assumption on the underlying data generation process, thus goes beyond pure data-driven methods.
> >
> > >  This difficulty is further compounded in off-policy and offline settings -> Why would that be?
> >
> > Thank you for pointing this out. We have explained the reasons in the revised manuscript, i.e., the gap between the (possibly unknown) behavior policy and the target policy, and unobserved confounders that influence both action and outcome.
> >
> > > The nodes corresponding to the exogenous variables are usually omitted in the causal graph. -> Why adopting the SCM framework then? Why not simply relying on causal graphs?
> >
> > Exogenous variables are typically omitted from causal graphs because they represent factors outside the model and can be summarized by probabilities, as explained by Pearl (2009b). This is also related to the causal Markov condition, which we have added to Section 2.1 in our revision. Causal graph-based methods implicitly assume the existence of an underlying SCM. Besides, causal graphs may not be sufficient to discuss counterfactuals while SCM provides a sound and complete mathematic framework (Pearl, 2009b).
> >
> > > all distributional shifts result [...] in the data generation proces
> >
> > Thank you for your comment. We respectfully disagree with your opinion. We have added references to support this statement (Schölkopf et al.,2021; Thams, 2022). The logic behind this statement is based on the fact that any change in distribution is always due to changes in at least one causal mechanism (Schölkopf et al., 2021), and intervention, by definition, implies changing causal mechanisms.
> >
> > Regarding the counterexample of post-hoc re-sampling, it implies the existence of a second-stage data generation process. In this process, the data from the first process is used as input, and the selection criteria define the structural equations. New selection criteria can be seen as different interventions in this process, resulting in changes to data distributions. Therefore, the statement remains true.
> >
> > > Figure 2 and Figure 3 cause confusion
> >
> > Thanks for the valuable comment. We have added explanations in the captions and the context. In Figure 2, the variables are binary. In Figure 3, X is a discrete variable with three possible values: fresh citrus fruit, heated juice, or rotten meat.
> >
> > > In particular, ... they overlap
> >
> > Thanks for the constructive comment. We have adjusted the statement to make it clearer. We have added explanations regarding the consistency assumption and the twin network method for illustrating counterfactuals.  When the evidence set is empty, a counterfactual query degenerates into an interventional one.

---

> > > ### Author Response · Authors · 2023-03-27
> > > **Detailed Response to Reviewer B1jj (2/n)**
> > >
> > > > Definition 2.2 seems to apply only to discrete action and state spaces.
> > >
> > > This is a generic definition that follows [1], applying to both infinite and finite MDPs.
> > >
> > > > the policy is not a causal relationship ... -> Why not?
> > >
> > > We appreciate the feedback. We have revised the statement and clarified that the policy is a soft intervention that reserves the dependence of the action on the state.
> > >
> > > > In general, changes ... -> How so? Why? Supporting evidence is missing.
> > >
> > > Thank you for pointing this out. We have updated the manuscript to address this concern by adding supporting references (Schölkopf et al., 2021; Perry et al., 2022) and clarifying this is a hypothesis based on the independent causal mechanisms principle .
> > >
> > > > Typos and minor adjustments.
> > >
> > > Thank you for your thoughtful comments. We have corrected the typos and made some adjustments based on your suggestions.
> > > - "Data alone cannot...": We have removed this statement from the introduction.
> > > - "RL agents equipped ...": We have removed this statement from the introduction and the difference w.r.t. regular MBRL is discussed in Section 3.1.2.
> > > - "the central role of causality ... ": We have adjusted the statement and added references.
> > > - "is an independent variable in ...": We have modified this term to "input variable".
> > > - "Definition 2.1 ...": We have refined the definition of SCM.
> > > - "while making changes ...": We have refined this statement about soft intervention.
> > > - "Causal factorization ... causal graph structure": We have removed this statement from the backgroud, and included a discussion in Section 3.2.2, clarifying that it is a hypothesis commonly used in prior work.
> > > - "Causal factorization [...] causal graph structure": We have removed this sentence as suggested.
> > > - "Figure 6: In the 3rd graph, Z is not a collider node.“: We have corrected the direction of arrows in Figure 6 (It's Figure 7 in the updated manuscript).
> > > - ”the observation may be infinite ... (finite but unobservable)”: We have clarified that the observation *space* may be infinite in a block MDP.
> > > - “Table 3: The text says that Gasse et al. (2021) ...“: We have added POMDP in Table 3.
> > > - "one of the most common ... do-calculus ...": We have revised the paragraph in Section 4.3 accordingly.
> > >
> > > [1] Agarwal, Alekh, et al. "Reinforcement learning: Theory and algorithms." _CS Dept., UW Seattle, Seattle, WA, USA, Tech. Rep_ (2019): 10-4.

---

### Review · Reviewer_9gmX · 2023-03-18

**Summary Of Contributions:**

This paper is a comprehensive survey of the emerging field of causal reinforcement learning (RL), which seeks to improve existing RL algorithms by utilizing structured and interpretable representations of the data generation process. The authors introduce the basic concepts of causality and RL and explain how causal modeling can address core challenges in non-causal RL. They categorize and systematically review existing causal RL approaches based on their target problems and methodologies. The paper highlights major unresolved issues and promising research directions in causal RL and proposes a problem-oriented taxonomy that will help researchers and practitioners better understand the advantages of causal modeling and identify solutions to the challenges they face. Overall, the paper establishes connections between existing work based on the structural causal model framework, and provides a clear and concise overview of the foundational concepts of causality research and RL.

**Audience:**

Yes

**Claims And Evidence:**

Yes

**Requested Changes:**

+ expand the discussion on connections to DTR and review the literature of DTR in the context of causal RL

+ ground the discussion of why causality helps RL in technical terms. e.g. show when causal inference helps and when it doesn't. I have found discussions in the following two papers clarifying and inspiring. Related technical ground of causality in causal RL will be essential to motivate and contextualize causal RL algorithms in later sections.

Chen, Yuansi, and Peter Bühlmann. "Domain adaptation under structural causal models." The Journal of Machine Learning Research 22, no. 1 (2021): 11856-11935.

Magliacane, Sara, Thijs Van Ommen, Tom Claassen, Stephan Bongers, Philip Versteeg, and Joris M. Mooij. "Domain adaptation by using causal inference to predict invariant conditional distributions." Advances in neural information processing systems 31 (2018).


**Strengths And Weaknesses:**

Strengths

+ Provides a broad range of works related to causality and RL, which could be useful to the community.

Weakness

+ The major issue I had with this paper is its lack of discussion of dynamic treatment regimes, which is a closely related problem with a huge literature developed in causal inference. As the paper mentioned offhand, dynamic treatment regimes can be modeled as an MDP with a global confounder. Despite this tight connection (separate names of closely related topics), the paper spends little (if any) effort discussing this connection and the DTR literature. It significantly compromises the utility of a survey, which connects and unifies related literature under the same umbrella. With such a close connection missing, it will also mislead future researchers to also ignore this connection to DTR.

+ I also find the discussion in Sec 3 vague and sometimes misleading. Why causality is important in RL is arguably one of the most important problem a survey in causal RL shall articulate. However, why causality help RL is discussed at a high level, with little concreteness of technical discussions that support these discussion. Many terms are also vague, e.g. "causal representation" etc. Articulating this motivation for causal inference shall be the most important stepping stone for the later sections.

---

> ### Author Response · Authors · 2023-03-27
> **Response to Reviewer 9gmX**
>
> Thank you for the encouraging review and the thoughtful feedback! Our paper aimed to establish connections between existing work in causal RL and provide a common ground for discussing this exciting field. We are glad you found the paper provides a clear and concise overview. Following your feedback, we added (1) extended discussion of dynamic treatment regimes, (2) technical disccision of the motivation with supporting references.
>
> Below is the detailed description of the additions we made to the paper based on your feedback. If there are any other improvements that can be made please let us know!
>
> > Expand the discussion on connections to DTR and review the literature of DTR in the context of causal RL.
>
> Thank you for this comment. We agree that DTR is an important and valuable research topic that is less familiar to most RL researchers. To highlight its significance, in the revised paper we added a paragraph to the end of Section 4.3.1, which briefly introduces DTR and its connection with causal RL, and review the literature in the context of causal RL.
>
> > Articulating the motivation for causal inference in section 3. ...  e.g., show when causal inference helps and when it doesn't.
>
> Thanks for this comment. We have revised Section 3 in our updated manuscript by including more technical terms, providing easy-to-understand examples, and adding supporting references. Our aim is to help readers understand the intuitions behind causal RL, including the limitations of non-causal methods and why causality may help address these issues. We have also added a discussion on why the terms used in existing literature are often vague and unclear to the end of Section 2.3. Finally, we have added a section on the limitations of causal RL in Section 4.5.

---

### Decision · Action_Editors · 2023-05-31

**Recommendation:** Reject

**Comment:**


As all reviewers agreed, the causal reinforcement learning (RL) is an important topic, a complete survey of causal RL will be beneficial to practitioners and researchers in the field, as well to newcomers.

During the discussion period, the authors successfully addressed several technical concerns raised by the reviewers.

However, the major concern among most of the reviewers still remains, as the reviewer B1jj and GZue summarized in the comments: the current version should be reorganized to emphasize the authors' attitute. Specifically, the current version summarizes the existing literature relatively comprehensive. But the current organization of the draft does not do a good job in ``reviewing papers that attempt to demonstrate positive or negative answers, and finally summarizing with a more evidence-based and nuanced answer (e.g., pinpoint the limitations and the remaining challenges)'' and the correction in the modified version is still not satisfied (quote from review B1jj).

After the discussion, all the reviewers eventually achieve consensus that the current version can be further improved for publishing.

**Audience:**


The survey is aiming for researchers and practicers in causal RL. I believe the community will be interested in this paper.

**Claims And Evidence:**


The major criticism from the reviewers is that the paper is not organized well, in the sense that the important open questions are not clearly stated, and the situation for each question is not claimed with enough evidence.

**Resubmission Of Major Revision:**

The authors may consider submitting a major revision at a later time.